# Quantitative insights into the cyanobacterial cell economy

**Tomáš Zavřel[1][†]\*, Marjan Faizi[2][†], Cristina Loureiro[3], Gereon Poschmann[4], Kai Stühler[4], Maria Sinetova[5], Anna Zorina[5], Ralf Steuer[2]\*, Jan Červený[1]**

[1]Laboratory of Adaptive Biotechnologies, Global Change Research Institute CAS, Brno, Czech Republic; [2]Institut für Biologie, Fachinstitut für Theoretische Biologie, Humboldt-Universität zu Berlin, Berlin, Germany; [3]Department of Applied Physics, Polytechnic University of Valencia, Valencia, Spain; [4]Molecular Proteomics Laboratory, BMFZ, Heinrich-Heine-Universität Düsseldorf, Düsseldorf, Germany; [5]Timiryazev Institute of Plant Physiology, Russian Academy of Sciences, Moscow, Russian Federation

**\*For correspondence:**
zavrel.t@czechglobe.cz (TZ);
ralf.steuer@hu-berlin.de (RS)

[†]These authors contributed equally to this work

**Competing interests:** The authors declare that no competing interests exist.

**Abstract** Phototrophic microorganisms are promising resources for green biotechnology. Compared to heterotrophic microorganisms, however, the cellular economy of phototrophic growth is still insufficiently understood. We provide a quantitative analysis of light-limited, light-saturated, and light-inhibited growth of the cyanobacterium *Synechocystis* sp. PCC 6803 using a reproducible cultivation setup. We report key physiological parameters, including growth rate, cell size, and photosynthetic activity over a wide range of light intensities. Intracellular proteins were quantified to monitor proteome allocation as a function of growth rate. Among other physiological acclimations, we identify an upregulation of the translational machinery and downregulation of light harvesting components with increasing light intensity and growth rate. The resulting growth laws are discussed in the context of a coarse-grained model of phototrophic growth and available data obtained by a comprehensive literature search. Our insights into quantitative aspects of cyanobacterial acclimations to different growth rates have implications to understand and optimize photosynthetic productivity.

DOI: https://doi.org/10.7554/eLife.42508.001

## Introduction

Cyanobacteria are key primary producers in many ecosystems and are an integral part of the global biogeochemical carbon and nitrogen cycles. Due to their fast growth rates, high productivity and amenability to genetic manipulations, cyanobacteria are considered as promising host organisms for synthesis of renewable bioproducts from atmospheric $CO_2$ (*Al-Haj et al., 2016*; *Zavřel et al., 2016*), and serve as important model organisms to understand and improve photosynthetic productivity.

Understanding the cellular limits of photosynthetic productivity in cyanobacteria, however, requires quantitative data about cellular physiology and growth: accurate accounting is central to understand the organization, growth and proliferation of cells (*Vázquez-Laslop and Mankin, 2014*). While quantitative insight into the cellular economy of phototrophic microorganisms is still scarce, the cellular economy of heterotrophic growth has been studied extensively—starting with the seminal works of Monod, Neidhardt, and others (*Neidhardt et al., 1990*; *Neidhardt, 1999*; *Jun et al., 2018*) to more recent quantitative studies of microbial resource allocation (*Molenaar et al., 2009*; *Klumpp et al., 2009*; *Scott et al., 2010*; *Scott and Hwa, 2011*; *Bosdriesz et al., 2015*; *Maitra and Dill, 2015*; *Weiße et al., 2015*). In response to changing environments, heterotrophic microorganisms are known to differentially allocate their resources: with increasing growth rate, heterotrophic microorganisms typically exhibit upregulation of ribosomes and other proteins related to translation

and protein synthesis (*Scott et al., 2010*; *Molenaar et al., 2009*; *Peebo et al., 2015*), exhibit complex changes in transcription profiles, for example (*Klumpp et al., 2009*; *Matsumoto et al., 2013*), and increase cell size (*Kafri et al., 2016*). The molecular limits of heterotrophic growth have been described thoroughly (*Kafri et al., 2016*; *Erickson et al., 2017*; *Scott et al., 2014*; *Metzl-Raz et al., 2017*; *Klumpp et al., 2013*).

In contrast, only few studies so far have addressed the limits of cyanobacterial growth from an experimental perspective (*Bernstein et al., 2016*; *Yu et al., 2015*; *Abernathy et al., 2017*; *Ungerer et al., 2018*; *Jahn et al., 2018*). Of particular interest were the acclimations that enable fast photoautotrophic growth (*Bernstein et al., 2016*; *Yu et al., 2015*; *Abernathy et al., 2017*; *Ungerer et al., 2018*). The cyanobacterium with the highest known photoautotrophic growth rate, growing with a doubling time of up to $T_D \sim 1.5\,\mathrm{h}$, is the strain *Synechococcus elongatus* UTEX 2973 (*Ungerer et al., 2018*). Compared to its closest relative, *Synechococcus elongatus* PCC 7942, the strain shows several physiological acclimations, such as higher PSI and cytochrome $b_6f$ content per cell (*Ungerer et al., 2018*), lower metabolite pool in central metabolism, less glycogen accumulation, and higher NADPH concentrations and higher energy charge (relative ATP ratio over ADP and AMP) (*Abernathy et al., 2017*). Recently, a study of the primary transcriptome of *Synechococcus elongatus* UTEX 2973 reported the increased transcription of genes associated with central metabolic pathways, repression of phycobilisome genes, and accelerated glycogen accumulation rates in high light compared to low light conditions (*Tan et al., 2018*).

While these studies point to strain-specific differences and are important for characterizing non-model microbial metabolism (*Abernathy et al., 2017*), the general principles of resource allocation in photoautotrophic metabolism and the laws of phototrophic growth are still poorly understood. Therefore, the aim of this study is to provide a consistent quantitative dataset of cyanobacterial physiology and protein abundance for a range of different light intensities and growth rates—and put the data into the context of published values obtained by a comprehensive literature search as well as into the context of a recent model of photosynthetic resource allocation (*Faizi et al., 2018*). To this end, we chose the widely used model strain *Synechocystis* sp. PCC 6803 (*Synechocystis* hereafter). Since *Synechocystis* exhibits significant variations with respect to both genotype (*Ikeuchi and Tabata, 2001*) and phenotype (*Morris et al., 2017*; *Zavřel et al., 2017*), we chose the substrain GT-L, a strain that has a documented stable phenotype for at least four years preceding this study. All data are obtained under highly reproducible and controlled experimental conditions, using flat-panel photobioreactors (*Nedbal et al., 2008*) within an identical setup as in the previous studies (*Zavřel et al., 2015b*).

The data obtained in this work provide a resource for quantitative insight into the allocation of cellular components during light-limited, light-saturated, and photoinhibited growth. In dependence of the light intensity and growth rate, we monitor key physiological properties, such as changes in cell size, dry weight, gas exchange (both $CO_2$ and $O_2$), as well as changes in abundance of pigments, DNA, total protein, and glycogen. Using proteomics, we show that ~57% (779 out of 1356 identified proteins) proteins changed their abundance in dependence of growth rate, whereas the rest was independent of growth rate. A detailed analysis of changes in individual protein fractions revealed phototrophic 'growth laws': abundances of proteins associated with light harvesting decreased with increasing light intensity and growth rate, whereas abundances of proteins associated with translation and biosynthesis increased with increasing light intensity and growth rate—which is in good agreement with recent computational models of cyanobacterial resource allocation (*Burnap, 2015*; *Rügen et al., 2015*; *Mueller et al., 2017*; *Reimers et al., 2017*; *Faizi et al., 2018*).

## Results

### Establishing a controlled and reproducible cultivation setup

The *Synechocystis* substrain GT-L (*Zavřel et al., 2015b*) was cultivated in flat panel photobioreactors (*Figure 1A*) using at least 5 independent reactors in a quasi-continuous (turbidostat) regime (*Figure 1B*), with red light intensities of $27.5 - 1100$ µmol(photons) m$^{-2}$s$^{-1}$, supplemented with a blue light intensity of 27.5 µmol(photons) m$^{-2}$s$^{-1}$. The addition of blue light avoids possible growth limitations in the absence of short wavelength photons (*Golden, 1995*). Steady-state specific growth rates in turbidostat mode were calculated from monitoring the optical density measured at a

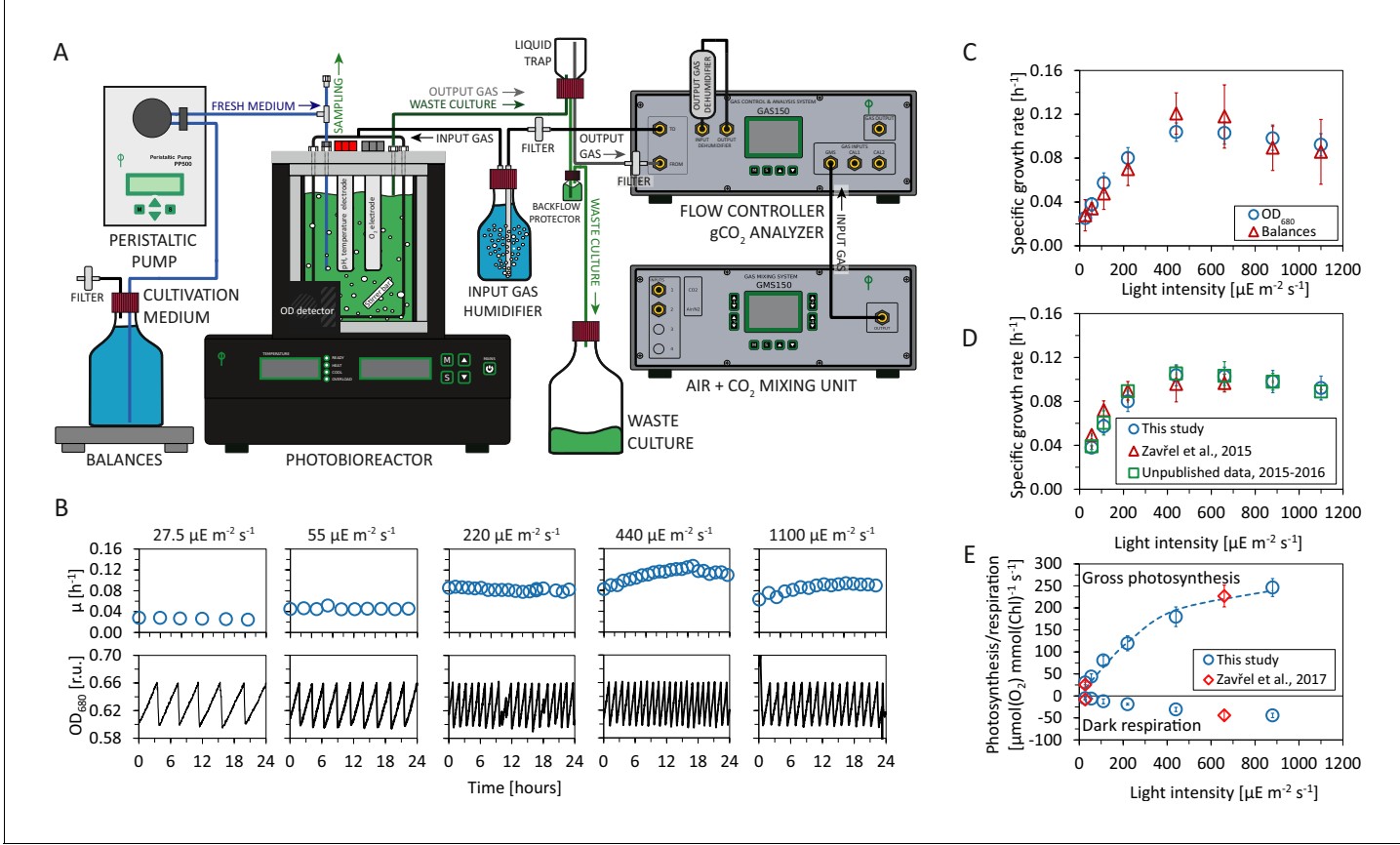

**Figure 1.** Experimental setup and evaluation of *Synechocystis* sp. PCC 6803 (substrain GT-L) phenotype stability. Panel A: Photobioreactor setup. Cultures were cultivated in a flat-panel photobioreactor vessel (400 mL) in a turbidostat regime according to *Zavřel et al. (2015b)*. Dilution of actively growing culture was based on measurements of optical density at 680 nm ($OD_{680}$). Inflow air and $CO_2$ were mixed in a gas mixing unit, the sparging gas flow rate was controlled by a gas analyzing unit. Sparging gas was moistened in a humidifier and, after bubbling through the photobioreactor vessel, separated from the waste culture via a liquid trap. $CO_2$ concentration in the output gas was measured by an infrared sensor according to *Červený et al., 2009*. All other parameters were set as described in *Nedbal et al. (2008)* and *Červený et al., 2009*. Panel B: Representative measurement of the $OD_{680}$ signal (black lines) within a turbidostat cultivation under increasing red light intensity (supplemented with low intensity of blue light). Calculation of specific growth rates (blue circles) is detailed in Materials and methods. Calculation of uptake and refilling rates of selected nutrients (including Na, (N, S, Ca, Mg, P and Fe) during the turbidostat cultivation is detailed in *Figure 1—source data 1* (the elemental composition of *Synechocystis* cells is based on data available in the literature). Panel C: Calculation of growth rates from the $OD_{680}$ signal and from top loading balances that monitored depletion rate of a spare cultivation medium (source data are available in *Figure 1—source data 2*). Panel D: Comparison of specific growth rates using an identical experimental setup throughout four successive years 2013–2017 (source data are available in *Figure 1—source data 3*). Panel E: Rates of gross photosynthesis and dark respiration, measured as $O_2$ evolution and consumption rates directly within the photobioreactor vessel throughout 5 min of light and dark periods in 2016–2017 (this study) and in 2015–2017 (*Zavřel et al., 2017*). The dashed line represents a P-I curve fit of data from this study according to *Platt et al. (1980)*. Source data are available in *Figure 1—source data 4*. *Figure 1C*: n = 6–11, *Figure 1D*: n = 3–11, *Figure 1E*: n = 4–6. Error bars (*Figure 1C–1E*) represent standard deviations.

DOI: https://doi.org/10.7554/eLife.42508.002

The following source data is available for figure 1:

**Source data 1.** Uptake and refilling rates of selected nutrients during the quasi-continuous cultivation.
DOI: https://doi.org/10.7554/eLife.42508.003
**Source data 2.** Source data for *Figure 1C*.
DOI: https://doi.org/10.7554/eLife.42508.004
**Source data 3.** Source data for *Figure 1D*.
DOI: https://doi.org/10.7554/eLife.42508.005
**Source data 4.** Source data for *Figure 1E*.
DOI: https://doi.org/10.7554/eLife.42508.006

wavelength of 680 nm ($OD_{680}$) as well as from the rate of depletion of spare cultivation medium (as measured by top loading balances). Both methods resulted in similar average values (*Figure 1C*). Estimation of the specific growth rates based on the medium depletion, however, exhibited higher variance. For further analysis, therefore, only values obtained from the $OD_{680}$ signal are reported.

The measured specific growth rates increased from $\mu = 0.025 \pm 0.002\,\mathrm{h}^{-1}$ to $\mu = 0.104 \pm 0.009\,\mathrm{h}^{-1}$ (corresponding to doubling times of $T_D \approx 27.7\mathrm{h} - 6.9\mathrm{h}$) with increasing light intensities up to 660 μmol(photons) m$^{-2}$s$^{-1}$ of red light. For higher light intensities the cultures exhibited photoinhibition—a reduction of the specific growth rate induced by high light intensities. Under the highest intensity of 1100 μmol(photons) m$^{-2}$s$^{-1}$, the specific growth rate decreased to $\mu = 0.093 \pm 0.011\,\mathrm{h}^{-1}$, corresponding to a doubling time of $T_D = 7.5\,\mathrm{h}$ (*Figure 1C–D*). The growth curve is consistent with previous measurements of cyanobacterial growth (*Zavřel et al., 2015b*; *Cordara et al., 2018*) and can be subdivided into three phases: light-limited, light-saturated, and photoinhibited growth.

The cultivation conditions, with (red) light intensity as the only variable, were highly controlled and reproducible. Temperature (32°C) and $CO_2$ concentration in the sparging gas (0.5%) were set to saturate *Synechocystis* growth in the exponential phase ($OD_{680} = 0.60 - 0.66$), as established in a previous study (*Zavřel et al., 2015b*). Refilling rate of selected nutrients (including Na, N, S, Ca, Mg, P and Fe) during the turbidostat cultivation was sufficient to prevent potential growth limitation by lack of any of these nutrients: see *Figure 1—source data 1* for further details (the elemental composition of *Synechocystis* cells considered for the calculations was based on data available in the literature).

The experimental setup, including the photobioreactor setup, light quality and intensity, temperature, composition of cultivation medium, $CO_2$ concentration in the sparging gas, bubbling and stirring rate was identical to the setup used in previous studies for this substrain (*Zavřel et al., 2015b*; *Zavřel et al., 2017*). We therefore could evaluate the stability of the *Synechocystis* sp. PCC 6803 GT-L phenotype throughout a four year period (2013–2017). *Figure 1D and E* show a comparison of the specific growth rates, as well as photosynthetic and respiration rates, from several previous studies (*Zavřel et al., 2015b*; *Zavřel et al., 2017*) and as yet unpublished data.

## Photosynthesis and respiration increase with light intensity and growth rate

The cultivation setup included a probe to monitor dissolved oxygen ($dO_2$) in the cultivation medium and a gas analyzing unit to measure $CO_2$ in the gas efflux. Online measurements of gas exchange rates allowed to assess dark respiration rates (measured as $O_2$ uptake rate during a 5 min dark period, see Materials and methods for further details) as well as photosynthetic activity (gross $O_2$ release rate and net $CO_2$ uptake rate). Both photosynthetic activity and dark respiration rates increased with increasing light intensity (*Figure 1E*, *Figure 2C–F*).

Between a light intensity of 27.5 and 880 μmol(photons) m$^{-2}$s$^{-1}$, the gross photosynthetic activity ($O_2$ release) increased from 30.5 ± 5.7 μmol($O_2$) mmol (Chl)$^{-1}$ s$^{-1}$ to 251.6 ± 49.4 μmol($O_2$) mmol (Chl)$^{-1}$ s$^{-1}$, and the dark respiration rate increased from 5.5 ± 2.7 μmol($O_2$) mmol (Chl)$^{-1}$ s$^{-1}$ to 40.9 ± 14.6 μmol($O_2$) mmol (Chl)$^{-1}$ s$^{-1}$ (*Figure 1E*).

Of particular interest were changes in gas exchange as a function of the specific growth rate. *Figure 2C–D* show gas exchange rates as a function of the specific growth rate per gram cellular dry weight (gDW), as well as per cell. Relative to gDW, $O_2$ release increased from 1.96 ± 0.691 mmol ($O_2$) gDW$^{-1}$ h$^{-1}$ to 5.92 ± 1.26 mmol ($O_2$) gDW$^{-1}$ h$^{-1}$ for an increase of growth rate from $\mu = 0.025 \pm 0.002\,\mathrm{h}^{-1}$ to $\mu = 0.099 \pm 0.013$ (*Figure 2C*). Dark respiration rate ($O_2$ consumption) increased from 0.35 ± 0.12 mmol ($O_2$) gDW$^{-1}$ h$^{-1}$ to 0.96 ± 0.21 mmol ($O_2$) gDW$^{-1}$ h$^{-1}$ (*Figure 2E–F*). Net $CO_2$ uptake rate increased from $0.78 \pm 0.37$ mmol ($CO_2$) gDW$^{-1}$ h$^{-1}$ to $4.01 \pm 0.50$ mmol ($CO_2$) gDW$^{-1}$ h$^{-1}$ (*Figure 2E*).

The measured gas exchange rates correspond to a photosynthesis:respiration (P:R) ratio (gross $O_2$ release relative to consumption) between $5.6 \pm 3.0$ and $7.5 \pm 2.5$. The photosynthetic quotient PQ (net $O_2$ release:$CO_2$ fixation) ranged from PQ = 2.1 ± 0.5 to PQ = 1.1 ± 0.4. The changes of both parameters (P:R and PQ) with respect to growth rate were not statistically significant (Kruskal-Wallis test: P:R ratio: $p - value = 0.88$, PQ: $p - value = 0.12$).

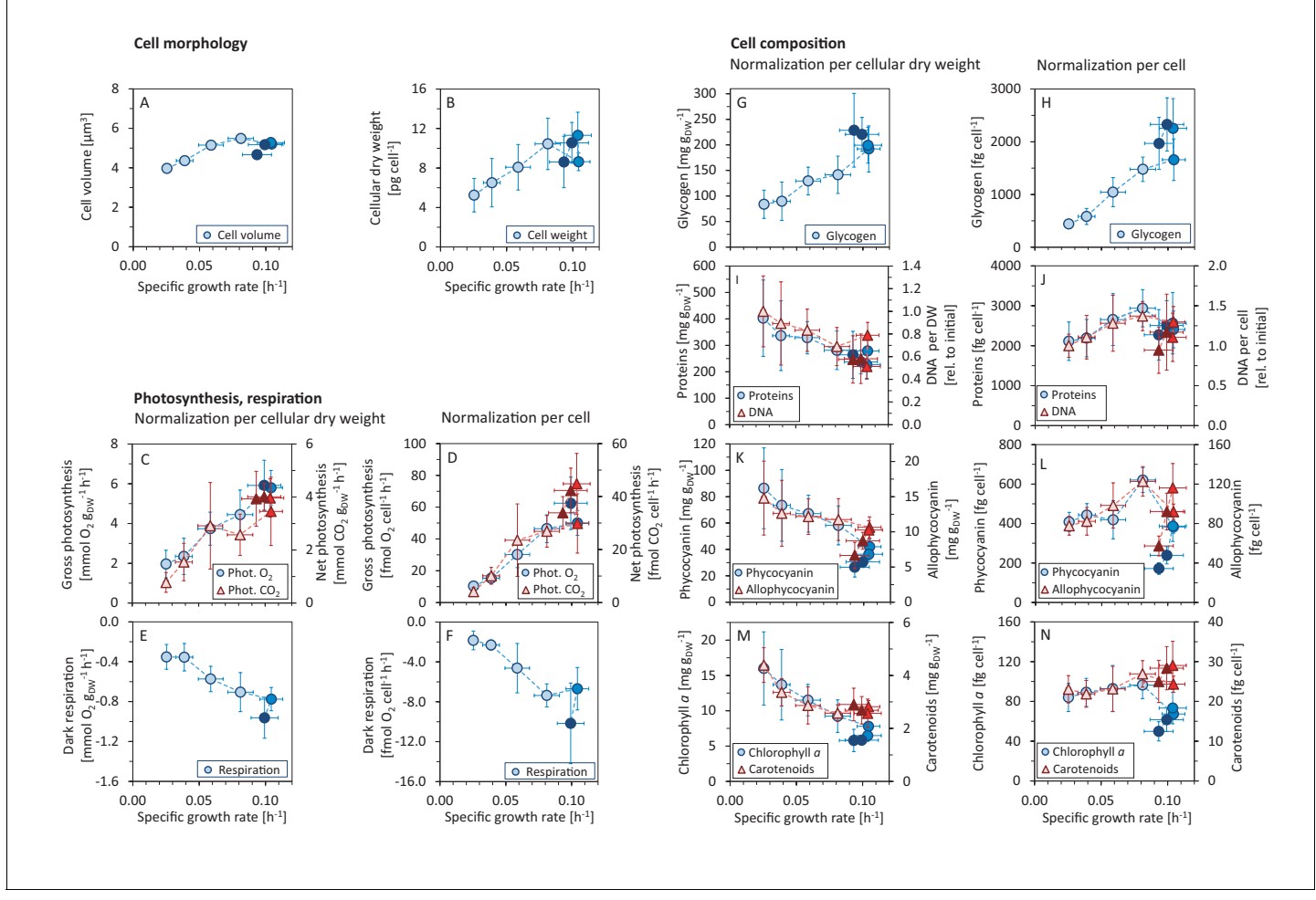

**Figure 2.** Variations in morphology and composition of *Synechocystis* cells with changing growth rate. Under increasing light intensity and changing growth rate, the following parameters were estimated: cellular volume (**A**) and dry weight (**B**), gross photosynthesis (**C, D**) and dark respiration (**E, F**), and content of glycogen (**G, H**), proteins, DNA (**I, J**), phycobiliproteins (**K, L**), chlorophyll *a* and carotenoids (**M, N**). The data are plotted relative to cellular dry weight (**C, E, G, I, K, M**) as well as per cell (**D, F, H, J, L, N**). DNA content was normalized to its initial value after standardization per dry weight and per cell, the measurement was only semi-quantitative. All values represent averages from 3 – 11 independent biological replicates, error bars represent standard deviations. If error bars are not visible (panel A), the standard deviation was too small for visualization. Within each figure, data points are displayed in three different color shades to reflect (from bright to dark) light-limited, light-saturated and light-inhibited growth. Data plotted as a function of light intensity are available in *Figure 2—figure supplement 1*. Comparison with data available in the literature is summarized in *Figure 2—source data 2*.

DOI: https://doi.org/10.7554/eLife.42508.007

The following source data and figure supplement are available for figure 2:

**Source data 1.** Source data for *Figure 2*.
DOI: https://doi.org/10.7554/eLife.42508.009
**Source data 2.** Comparison of the values measured in this study with data reported in the literature.
DOI: https://doi.org/10.7554/eLife.42508.010
**Figure supplement 1.** Allocation of key cellular resources as a function of light intensity.
DOI: https://doi.org/10.7554/eLife.42508.008

## Cell morphology and composition acclimate to changes in light intensity and growth rate

Culture samples were harvested under different light intensities to investigate the allocation of key cellular components as a function of growth rate. Cellular parameters included cell count, cell size, cell dry mass, as well as concentrations of glycogen, total protein, total DNA, phycocyanin, allophycocyanin, chlorophyll *a*, and carotenoids. The results (data normalized per gDW as well as per cell)

are summarized in *Figure 2* as a function of the specific growth rate; the results as a function of light intensity are summarized in *Figure 2—figure supplement 1*.

With increasing growth rate, the volume and weight of *Synechocystis* cells increased (*Figure 2A–B*). The cell diameter increased from $1.96 \pm 0.03 \, \mu m$ to $2.19 \pm 0.03 \, \mu m$, and slightly decreased again under photoinhibition. Since *Synechocystis* has a spherical cell shape, the estimated diameters correspond to cell volumes ranging from 3.97 μm$^3$ to 5.49 μm$^3$ (*Figure 2A*). Changes in cell volume were reflected in changes in cellular dry weight. Dry weight per cell increased from $5.3 \pm 1.7$ pg cell$^{-1}$ for the slowest specific growth rate to $11.3 \pm 2.3$ pg cell$^{-1}$ at the maximal growth rate. Under photoinhibition, cellular dry weight again decreased to $8.6 \pm 2.6$ pg cell$^{-1}$ (*Figure 2B*, *Figure 2—figure supplement 1*). The ratio of cellular dry weight to cell volume showed no significant change for different growth rates (Kruskal-Wallis test: $p - value = 0.077$).

The amount of glycogen per gDW increased with increasing growth rate, from $84 \pm 28$ mg gDW$^{-1}$ to $199 \pm 35$ mg gDW$^{-1}$ for the maximal growth rate, and further increased to $229 \pm 72$ mg gDW$^{-1}$ under conditions of photoinhibition (*Figure 2G*). These values correspond to an increase of glycogen per cell from $440 \pm 79$ fg cell$^{-1}$ to $2329 \pm 504$ fg cell$^{-1}$ (*Figure 2H*).

In contrast, the protein content per gDW decreased with increasing growth rate. Protein content per cell, however, did not change significantly for different light intensities and growth rates (Kruskal-Wallis test: $p - value = 0.076$). The absolute values of protein content were between $402 \pm 144$ and $227 \pm 6$ mg gDW$^{-1}$ (*Figure 2I*), and between $2144 \pm 482$ and $2937 \pm 466$ fg cell$^{-1}$ (*Figure 2J*).

Changes in DNA content were only estimated in relative units and are reported relative to the DNA content at the lowest growth rate. With increasing growth rate, the DNA content normalized per gDW decreased to $51 \pm 11\%$ of the initial value (*Figure 2I*). The (relative) DNA content per cell, however, increased with increasing growth rate up to $137 \pm 19\%$ of its initial value. Under conditions of photoinhibition, the relative DNA content per cell decreased again to $94 \pm 29\%$ of the initial value (*Figure 2J*).

Relative to gDW, the amounts of phycobiliproteins, chlorophyll *a* and carotenoids decreased with increasing growth rate. Under conditions of photoinhibition, we observed additional reduction of these pigments per gDW (*Figure 2K,M*). When considering the concentrations per cell, however, the respective amounts initially increased with increasing growth rates, and decreased again under conditions of photoinhibition. Overall, pigment content decreased with increasing light intensity (irrespective of normalization), with the exception of carotenoids that exhibited a slight increase per cell as a function of light intensity. The changes of pigment amounts as a function of growth rate (relative to gDW as well as per cell) were significant (Kruskal-Wallis test: $p - value < 0.05$, see Materials and methods for further details). The absolute amounts of phycocyanin were between $86.4 \pm 30.7$ and $26.5 \pm 7.5$ mg gDW$^{-1}$, corresponding to $172 \pm 29$ and $620 \pm 63$ fg cell$^{-1}$ (*Figure 2K,L*), the amounts of allophycocyanin were between $14.8 \pm 5.3$ and $6.7 \pm 1.9$ mg gDW$^{-1}$, corresponding to $57 \pm 10$ and $123 \pm 15$ fg cell$^{-1}$ (*Figure 2K,L*). The absolute amounts of chlorophyll *a* were between $16 \pm 5.2$ and $5.8 \pm 1.6$ mg gDW$^{-1}$, corresponding to a range between $50 \pm 10$ and $96 \pm 14$ fg cell$^{-1}$ (*Figure 2M,N*), the absolute amounts of carotenoids were between $4.4 \pm 0.7$ and $2.6 \pm 0.5$ mg gDW$^{-1}$, corresponding to a range between $22 \pm 3$ and $29 \pm 6$ fg cell$^{-1}$ (*Figure 2M,N*).

To put the data into context, we conducted a comprehensive literature research with respect to reported physiological parameters of *Synechocystis* sp. PCC 6803. The results are summarized in *Figure 2—source data 2*, and the data include also meta information on experimental conditions. Overall, the values obtained in this study are in good agreement with the previously reported values. Individual parameters, however, exhibit high variability due to the wide range of different experimental conditions.

## Proteome allocation as a function of growth rate

Culture samples for 6 light intensities were harvested to obtain quantitative proteome profiles using mass spectrometry, with 5 biological replicates for each light intensity. We chose a label-free quantification (LFQ) approach to access relative and absolute protein amounts. Here, the peptide precursor ion intensities (MS1) were used for protein quantification. The results of the proteomics analysis are summarized in *Figure 3*. We identified 1356 proteins (the complete list is provided in *Figure 3—source data 1*). Of these, the (relative) abundances of 779 proteins (57%) significantly changed with growth rate (Kruskal-Wallis test: $p - value < 0.05$), the (relative) abundances of the remaining 577 proteins (43%) were independent of growth rate. We obtained functional annotation for all 1356

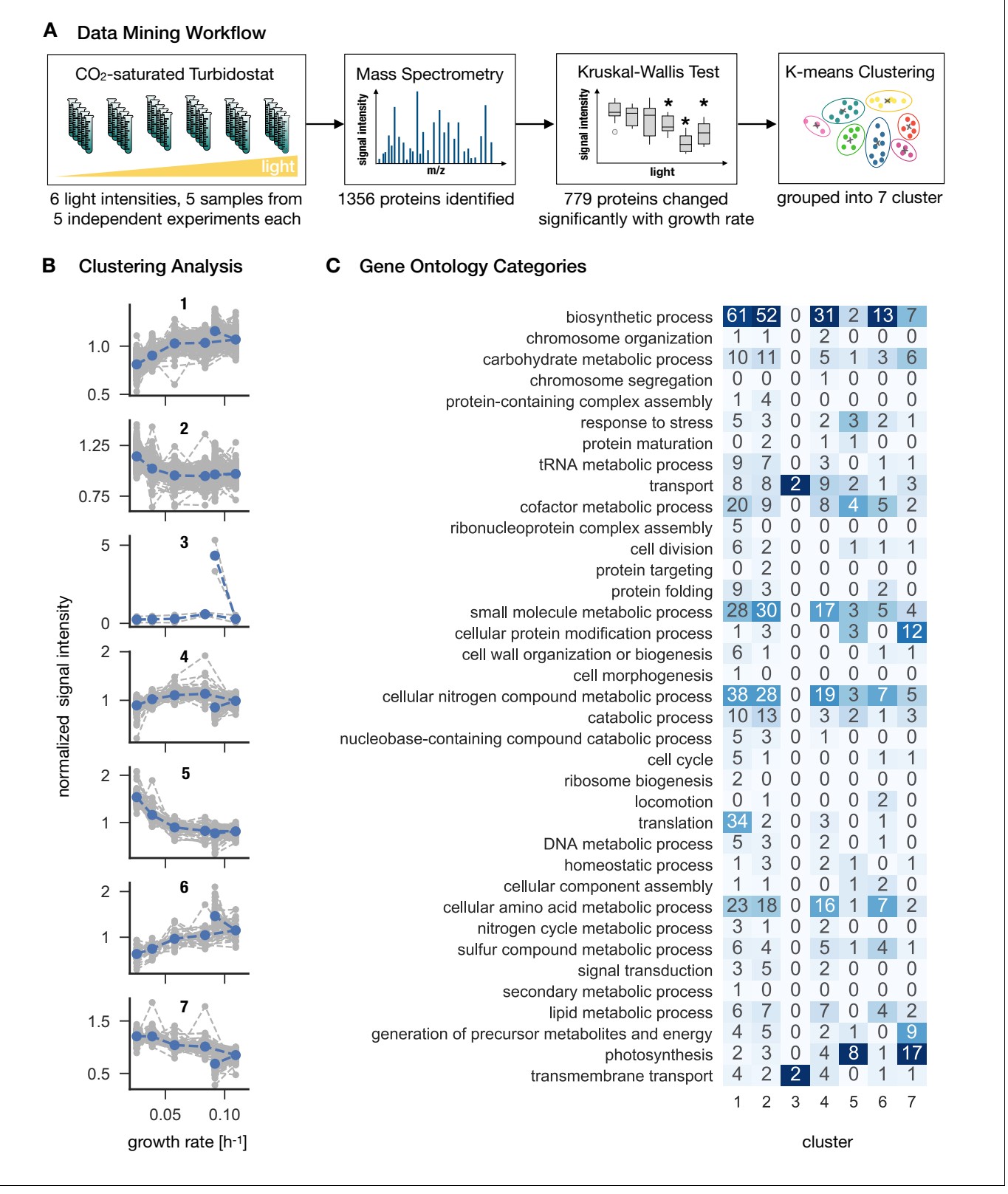

**Figure 3.** *Synechocystis* proteome allocation as a function of growth rate. Panel A: The workflow. Samples were harvested and analyzed by mass spectrometry (the proteomics dataset is available in *Figure 3—source data 1*). A Kruskal-Wallis test was used to distinguish between growth-dependent and growth-independent proteins. 779 growth-dependent and 577 growth-independent proteins were identified. Panel B: Clustering analysis. Based on k-means clustering analysis (*Figure 3—figure supplement 1*), the 779 growth-dependent proteins were separated into seven
*Figure 3 continued on next page*

*Figure 3 continued*

clusters. Gray dashed lines represent protein abundances as medians of 5 biological replicates, normalized by the respective means. Blue dashed lines represent centroids of the respective clusters. Panel C: Proteins were annotated using the GO classes, the matrix represents the annotation mapped to GO slim categories. Proteins can be associated to several GO slim categories. The highest ranking annotation per cluster is highlighted in dark blue.

DOI: https://doi.org/10.7554/eLife.42508.011

The following source data and figure supplement are available for figure 3:

**Source data 1.** Proteomics dataset.
DOI: https://doi.org/10.7554/eLife.42508.013
**Source data 2.** List of growth-dependent proteins.
DOI: https://doi.org/10.7554/eLife.42508.014
**Source data 3.** List of growth-independent proteins.
DOI: https://doi.org/10.7554/eLife.42508.015
**Figure supplement 1.** Elbow method for the identification of an appropriate number of clusters (grey dashed line at seven clusters).
DOI: https://doi.org/10.7554/eLife.42508.012

proteins using the Gene Ontology (GO) database (*Ashburner et al., 2000*). Of the 779 growth-dependent proteins, 450 were annotated with non-trivial categories (excluding categories such as *unknown* or *putative*), of the 577 growth-independent proteins, 303 were annotated with non-trivial categories. To facilitate the analysis, the functional annotation was mapped to a subset GO slim (higher level GO terms, (*Klopfenstein et al., 2018*)), which resulted in 40 distinct GO terms (each protein might be associated with more than one annotation). Significant differences (Fisher's exact test, $p - value < 0.05$) between growth-dependent and growth-independent annotations are summarized in *Table 1*. Growth-dependent proteins exhibited an over-representation of categories such as *Translation*, *Protein folding*, *Cell division* and *Photosynthesis*, among others.

To allow for a more detailed analysis of growth-dependent proteins, the changes in abundance of the 779 proteins were grouped into 7 clusters using k-means clustering (*Figure 3—figure supplement 1*). The number of clusters was determined using the elbow method. The identified clusters corresponded either to upregulation (cluster 1 and 6), or downregulation of protein abundance with growth rate (cluster 2, 5, 7) or more complex changes (cluster 3 and 4). The results of the clustering analysis are summarized in *Figure 3*, along with an annotation matrix that highlights the prevalent function (GO slim) categories for each cluster. The growth-dependent proteins encompass 37 distinct annotations mapped to GO slim categories.

**Table 1.** Gene Ontology (GO) slim categories (*Klopfenstein et al., 2018*) with the amount of associated growth-dependent and independent proteins.

A complete list of the GO slim categories is provided in *Table 1—source data 1*. Here, only categories that exhibit a significant difference (Fisher's exact test, $p - value < 0.05$) between growth-dependent and independent groups are listed. Shown is the number of annotations per category.

| Gene ontology categories | Growth dependent | Growth independent |
|---|---|---|
| Translation | 40 | 13 |
| Transport | 36 | 14 |
| Photosynthesis | 36 | 8 |
| Catabolic process | 32 | 4 |
| Protein folding | 14 | 3 |
| Cell division | 12 | 0 |
| Cell wall organization or biogenesis | 10 | 1 |
| Cell cycle | 9 | 0 |

DOI: https://doi.org/10.7554/eLife.42508.016

The following source data is available for Table 1:

**Source data 1.** List of all 40 GO slim categories with the respective amounts of growth-dependent and growth-independent proteins (and their cluster associations).
DOI: https://doi.org/10.7554/eLife.42508.017

Cluster 1 (192 proteins) and 6 (41 proteins) exhibit increasing abundance for increasing light intensity and growth rate. Prevalent annotations are *biosynthetic processes*, such as *cellular nitrogen compound metabolic processes*, *cellular amino acid metabolic processes*, as well as, for cluster 1, *translation*. Cluster with low variation (Cluster 2, 218 proteins) and cluster with ambiguous shapes (cluster 4, 124 proteins) exhibit a similar set of categories as cluster 1 and 6. In contrast, both clusters that exhibit a clear decrease with increasing light intensity and growth rate (cluster 5, 65 proteins and cluster 7, 79 proteins) are both annotated with *photosynthesis* as the highest-ranking annotation. Finally, cluster 3 (2 proteins) exhibits a sharp upregulation during photoinhibition, with both proteins annotated with the categories *transport* and *transmembrane processes*.

We note that, similar to some of the physiological properties as shown in *Figure 2*, the abundances of clusters 1, 3, 4, 6 and 7 exhibited a characteristic 'kink' at high growth rates corresponding to a sharp up- or downregulation under photoinhibition (*Figure 3B*).

## Visualization of functional annotation using proteomaps

To complement the clustering analysis, we used the proteomaps software (www.proteomaps.net; *Liebermeister et al., 2014*) to visualize the relative abundances of the identified proteins for different light conditions. To this end, iBAQ intensities were used as an approximation for quantitative protein amounts. Here, the measured precursor ion intensities (MS1) for each individual protein are summed up and divided by the number of theoretically observable peptides for the respective protein. The number of theoretically observable peptides is calculated for each protein by an in silico digestion of the respective database sequence and only peptides between 6 and 30 amino acids in length are considered for the calculations. We emphasize that, while iBAQ intensities are roughly proportional to the molar amounts of the proteins, iBAQ intensities only refer to identified proteins and do not reflect the whole proteome: the sum of all proteins used for the generation of proteomaps is based on identified proteins only, with the unidentified proteins being neglected. Therefore, the proportionality factor could change from sample to sample, and the intensities are interpreted only as approximations that provide insight into the expected overall abundances.

*Figure 4* shows proteomaps for three distinct growth regimes: light-limited growth at 27.5 µmol (photons) m$^{-2}$s$^{-1}$ (specific growth rate $\mu = 0.025\,\mathrm{h}^{-1}$), light-saturated growth at 440 µmol(photons) m$^{-2}$s$^{-1}$ (specific growth rate $\mu = 0.104\,\mathrm{h}^{-1}$), and photoinhibited growth at 1100 µmol(photons) m$^{-2}$s$^{-1}$ (specific growth rate $\mu = 0.093\,\mathrm{h}^{-1}$). The full set of proteomaps is available in *Figure 4—figure supplement 1*.

The proteomaps (annotated using Cyanobase (*Fujisawa et al., 2017*) mapped to custom KEGG annotation) show similar trends as the clustering analysis: upregulation of proteins associated with translational processes and ribosomes with increasing light intensity and growth rate, and downregulation of photosynthetic and light harvesting proteins with increasing light intensity and growth rate.

## A coarse-grained model provides insight into proteome allocation

To interpret the experimental results on cyanobacterial physiology, we made use of a semi-quantitative resource allocation model of cyanobacterial phototrophic growth. The model was adopted from *Faizi et al. (2018)* and is summarized in *Figure 5*. In brief, the model includes coarse-grained proteome fractions for cellular processes related to growth, including carbon uptake $T$, metabolism $M$, photosynthesis $P$, and ribosomes $R$. The model describes light-dependent cyanobacterial growth at saturating conditions of external inorganic carbon. Compared to the original model from *Faizi et al. (2018)*, we now included a growth-independent protein fraction $Q$ that accounts for half of the proteome mass. All further (minor) modifications and changes in the model definition are detailed in Materials and methods.

Following *Faizi et al. (2018)*, all kinetic parameters were sourced from the primary literature, except the parameters for the photosynthetic cross section, photosynthetic turnover rate, and the rate constant for photoinhibition (see Materials and methods for further details). These 3 parameters were fitted numerically, such that the predicted maximal growth rate µ (*Figure 1C–D*) matched the experimental values (*Figure 5B*). The stoichiometry and energy requirements for biosynthesis were approximated using a genome-scale model (*Knoop et al., 2013*). No proteomics data were used during model parametrization and fitting. All parameters and model definitions are provided in *Supplementary file 1*.

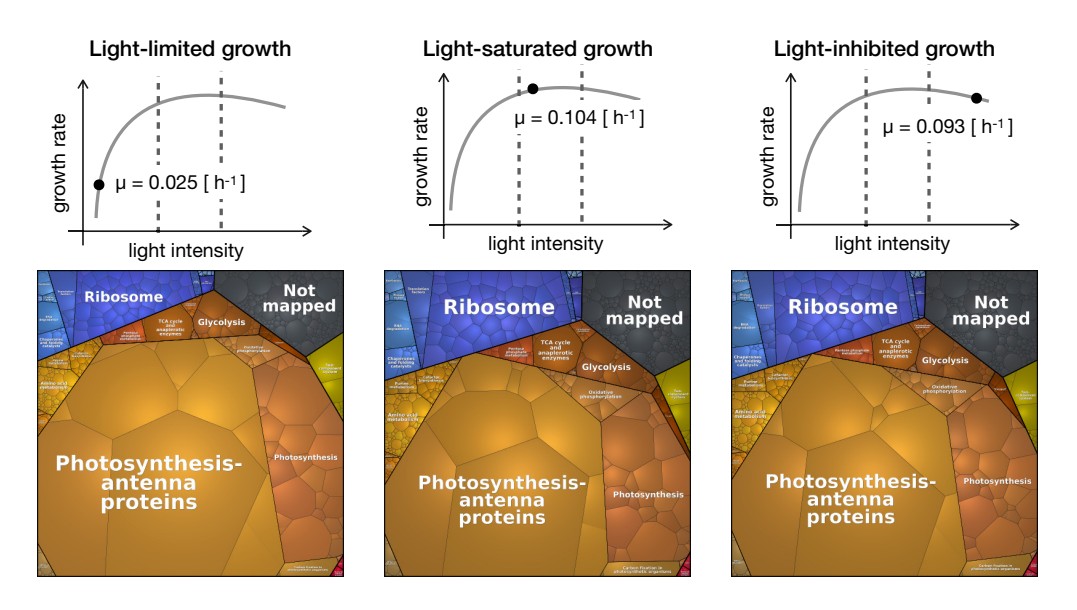

**Figure 4.** Proteomaps of proteome reallocation in *Synechocystis* under light-limited (27.5 µmol(photons) m$^{-2}$s$^{-1}$), light-saturated (440 µmol(photons) m$^{-2}$s$^{-1}$) and photoinhibited growth (1100 µmol(photons) m$^{-2}$s$^{-1}$).

DOI: https://doi.org/10.7554/eLife.42508.018

The following figure supplement is available for figure 4:

**Figure supplement 1.** Proteomaps of levels 2, 3 and 4.

DOI: https://doi.org/10.7554/eLife.42508.019

Evaluation of the model is based on the assumption of (evolutionary) optimality. That is, the model is solved using an optimization algorithm that maximizes the specific growth rate µ as a function of protein allocation. In this way, the model is able to predict how the coarse-grained proteome fractions are optimally allocated with increasing light intensity (*Figure 5B*). These predictions provide a reference to which the experimental data can be compared. We emphasize that such a comparison does not presuppose that proteome allocation in *Synechocystis* is necessarily optimal.

The model predictions are shown in *Figure 6*, together with data from the experimental analysis. The model predicts that the protein fraction associated with biosynthesis ($M$), as well as the ribosomal fraction ($R$), increases with increasing growth rate, in accordance with known growth laws of heterotrophic growth (*Scott et al., 2010*; *Weiße et al., 2015*). In contrast, the predicted protein fraction associated with photosynthesis ($P$, light harvesting and photosystems) decreases with increasing light intensity and growth rate. We highlight that the predicted growth laws exhibit a characteristic 'kink' under conditions of photoinhibition—a feature that is different from all reported growth laws for heterotrophic growth. Model predictions for the ribosomal and photosynthetic protein fractions are in good agreement with changes observed in the proteomics data (*Figure 6B*), whereas the experimentally observed fraction of the proteome assigned to metabolic functions (using the metabolic reconstruction of *Knoop et al., 2013*) exhibits no significant change, in contrast to model predictions.

## Testing protein allocation using immunoblotting analysis

In addition to large-scale proteomics, we tested the changes of selected proteins as a function of growth rate using immunoblotting analysis. Specifically, we measured the abundances of PsaC (an essential component of PSI), PsbA (the D1 protein of PSII), the RuBisCO subunit RbcL, and the ribosomal proteins S1 and L1 under increasing growth rate. Additionally, the absolute amounts of PsbA, PsaC, and RbcL proteins were estimated by serial dilution of protein standards (see Materials and methods for details).

The immunoblotting results are summarized in *Figure 6C*, together with the model predictions and selected proteomics data. Overall, the trends confirm the results of the previous sections—and

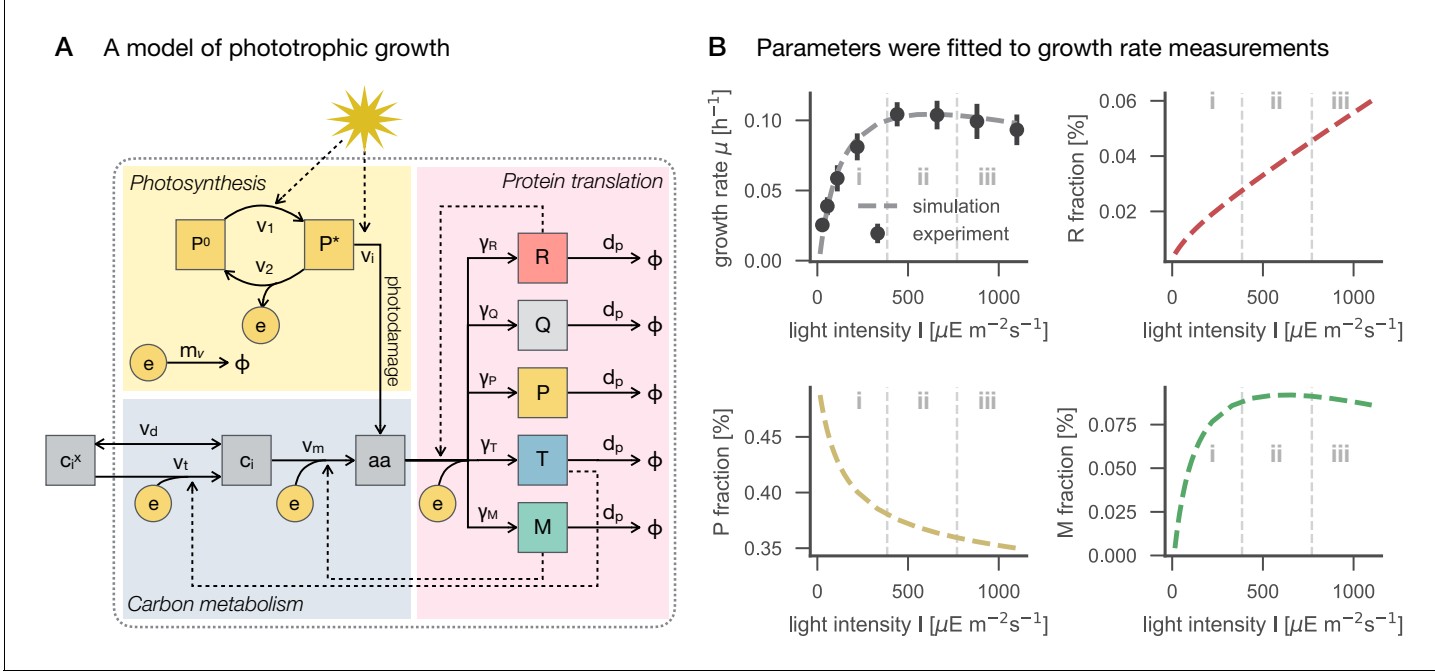

**Figure 5.** A model of phototrophic growth and reproduction of experimental growth curves. Panel A: A coarse-grained model of phototrophic growth, adopted from *Faizi et al. (2018)*. The model describes optimal proteome allocation under conditions of (i) light-limited, (ii) light-saturated and (iii) light-inhibited growth. Coarse-grained cellular processes include passive ($v_d$) and active import ($v_t$) of external inorganic carbon $c_i^x$, conversion of inorganic carbon $c_i$ into amino acids $aa$ ($v_m$), light harvesting and provision of cellular energy by photosynthesis ($v_1$ and $v_2$), as well as maintenance and photodamage ($m_v$ and $v_i$). Amino acids are translated into coarse-grained protein fractions for transport ($T$), metabolism ($M$), ribosomes ($R$), photosynthetic electron transport ($P$), as well as a growth-independent proteome fraction $Q$. Translation is limited by the amount of available ribosomes $R$. Panel B: The model reproduces the measured growth curve (*Figure 1C–D*) as a function of light intensity. Shown are the specific growth rate μ, as well as the main proteome fractions predicted by the model: ribosome ($R$) fraction, photosynthetic electron transport ($P$) fraction, and metabolism ($M$) fraction, as a function of light intensity.

DOI: https://doi.org/10.7554/eLife.42508.020

correspond to the changes obtained from the protein allocation model. The ribosomal proteins S1 and L1 increased with increasing growth rate, with a characteristic upwards 'kink' under photoinhibition. The relative amount of PsbA, the D1 protein of PSII, decreased with increasing growth rate, with a characteristic downward 'kink' under photoinhibition (albeit less pronounced than for ribosomal proteins). PsaC associated to PSI followed a similar trend but with high variance. In contrast to the overall behavior of proteins associated with metabolism, the RuBisCO subunit RbcL exhibited a (slight) increase for increasing growth rates, in accordance with the model predictions (*Figure 6C*).

## Quantitative evaluation of selected protein complexes

Using the combined data of iBAQ intensities and quantification by immunoblotting and mass spectrometry, allows us to provide estimates of absolute amounts of selected protein complexes in *Synechocystis* cells. The results are summarized in *Table 2*, details of the calculations are listed in *Table 2—Source data 1*.

The most abundant proteins in *Synechocystis* cells were proteins associated to photosynthesis and carbon fixation, in particular proteins related to phycobilisomes, photosystems and RuBisCO. Aside from protein complexes, the most abundant monomeric protein was the elongation factor Tu (TufA) with approximately $2 - 3 \cdot 10^5$ copies per cell. Abundances of photosynthetic proteins were generally one to two orders of magnitude lower, similar to ribosomal and other proteins, including phosphoglycerate kinase, transketolase, PII signal transducing protein, ferredoxin-NADP reductase, D-fructose 1,6-bisphosphatase, glyceraldehyde-3-phosphate dehydrogenase, plastocyanin, superoxide dismutase, orange carotenoid protein, RNA polymerase, cytochrome $b_6f$ and chaperonine GroEL.

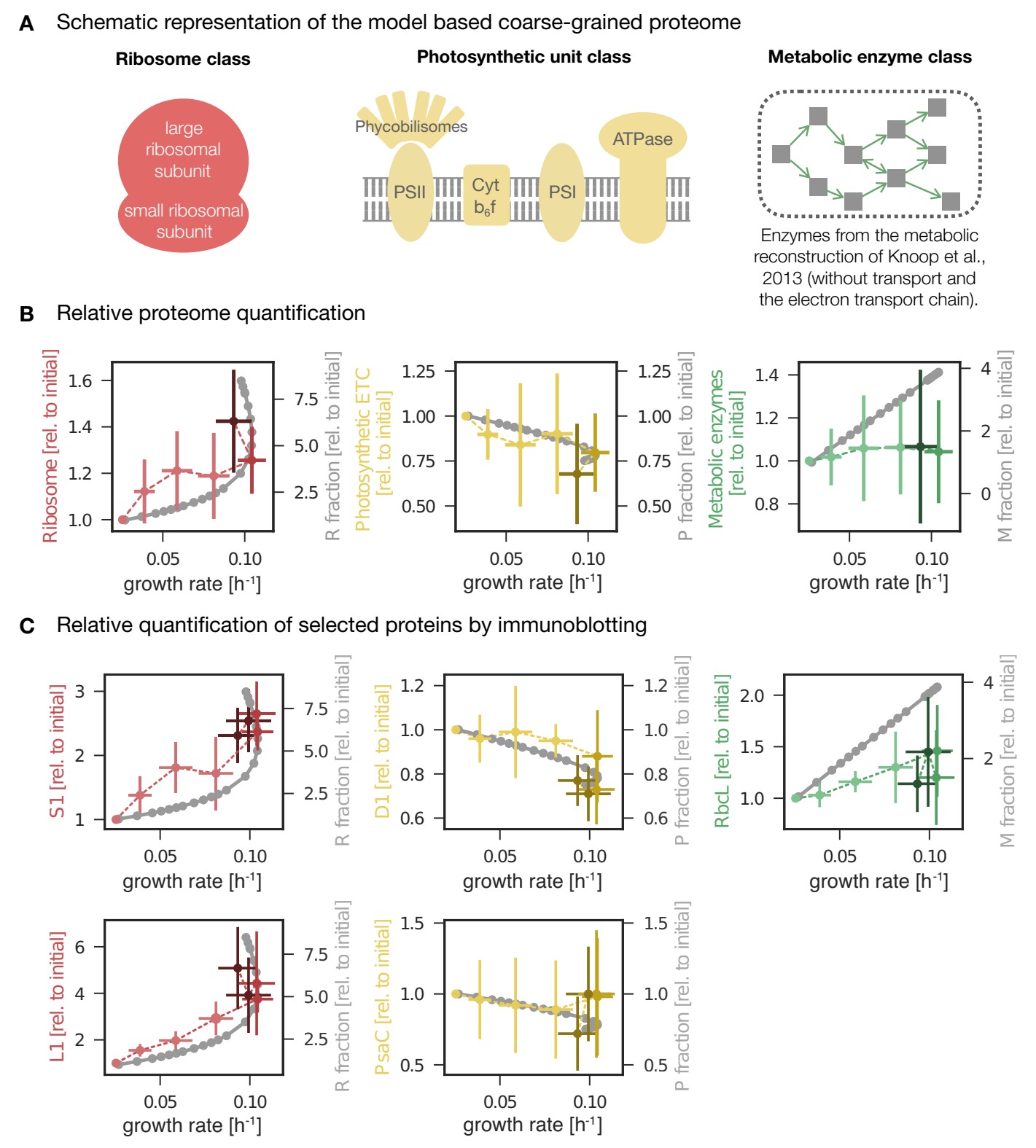

**Figure 6.** Changes in protein abundance as a function of specific growth rate compared to the predictions obtained from a computational model of proteome allocation. Panel A: Schematic representation of ribosome, photosynthetic units and metabolic enzyme classes considered in the proteome allocation model. Panel B: Relative proteomics data ( LFQ, label-free quantification intensities, left axes, mean fold change ± SD) of protein classes in comparison with the model predictions (grey lines, right axes). Panel C: Relative protein abundances obtained by immunoblotting analysis for selected

*Figure 6 continued on next page*

*Figure 6 continued*
proteins (left axes, median fold change ± SD) in comparison with coarse-grained model predictions (grey lines, right axes). Experimental values represent averages from 5 independent experiments, the error bars represent standard deviations. Panels B-C: The experimental data points are displayed in three different color shading to reflect (from bright to dark) light-limited, light-saturated and light-inhibited growth. The full dataset of the immunoblotting analysis is provided in *Figure 6—source data 1* and *Figure 6—figure supplement 1*. The list of proteins considered for ribosome, photosynthetic unit and metabolic enzyme classes is listed in *Figure 6—source data 2*. The influence of constant ribosomal, photosynthetic unit and metabolic enzyme classes on cellular growth rate is simulated in *Figure 6—figure supplement 2*.
DOI: https://doi.org/10.7554/eLife.42508.021

The following source data and figure supplements are available for figure 6:

**Source data 1.** Results of the immunoblotting analysis.
DOI: https://doi.org/10.7554/eLife.42508.024

**Source data 2.** List of proteins considered for ribosome, photosynthetic unit and metabolic enzyme classes.
DOI: https://doi.org/10.7554/eLife.42508.025

**Figure supplement 1.** Immunoblots and a list of antibodies used for the immunoblotting analysis.
DOI: https://doi.org/10.7554/eLife.42508.022

**Figure supplement 2.** Model simulations for investigating the influence of constant enzyme fractions on the cellular growth rate.
DOI: https://doi.org/10.7554/eLife.42508.023

*Table 2* also includes several previous estimates of protein abundances. We note that a direct comparison is challenging due to differences in cultivation conditions, including type of cultivation and cultivation vessel, cultivation media, irradiance, temperature, aeration, pH and the particular *Synechocystis* substrain (see *Figure 2—source data 2* for further details on particular experimental conditions).

## Discussion

### Quantitative resource allocation in cyanobacteria

Cyanobacteria are increasingly important host organisms for green biotechnology, but as yet insight into resource allocation of these organisms is restricted to few studies (*Abernathy et al., 2017*; *Burnap, 2015*; *Faizi et al., 2018*; *Jahn et al., 2018*). The scarcity of data is partially due to the fact that a quantitative experimental assessment of phototrophic growth is subject to a number of technical difficulties and standardized cultivation conditions are not available. The diversity of culture conditions used in the literature (summarized in *Figure 2—source data 2*) makes a direct comparison of the literature data difficult and often key parameters, such as specific growth rate, spectral properties of the light source, vessel geometry or gas exchange rates are not reported in sufficient detail. The premise of this study was therefore to use a highly reproducible cultivation setup that enables stable culture conditions in turbidostat mode and to provide a broad characterization of physiological parameters that can be compared to reported literature values. The results, interpreted in the context of a coarse-grained computational model of cyanobacterial resource allocation, provide further understanding of resource allocation and the cellular protein economy during light-limited, light-saturated and light-inhibited cyanobacterial growth.

### Maximal growth rates and glycogen accumulation

The maximal specific growth rates of *Synechocystis* GT-L obtained in this study (*Figure 1C,D*) were similar to the maximal growth rates of other *Synechocystis* substrains reported in previous studies (*Touloupakis et al., 2015*; *Nguyen and Rittmann, 2016*; *Du et al., 2016*; *Jahn et al., 2018*). While individual *Synechocystis* substrains can be more sensitive to high light (*Zavřel et al., 2017*), the agreement with previously reported values suggests an upper limit of *Synechocystis* growth in buffered BG-11 medium. However, (*van Alphen et al., 2018*) recently reported a specific growth rate of $0.16\,\mathrm{h}^{-1}$ ($T_D = 4.3\,\mathrm{h}$) using BG-11 medium with modified iron source and chelating agents. This finding suggests that the standard composition of BG-11 medium still induces a growth limitation, even though in our study the total concentration of iron and other elements refilled during the turbidostat cultivation was sufficient to fully saturate *Synechocystis* growth (*Figure 1—source data 1*).

**Table 2.** Quantification of selected protein complexes in *Synechocystis* cells.

Protein abundances were estimated as molecules per cell, as inferred from mass spectrometry, immunoblotting and spectrophotometric analysis. The stoichiometries of protein complexes were based on Uniprot (www.uniprot.org, (**UniProt Consortium, 2018**)) and RCSB (www.rcsb.org, (**Berman, 2000**)) databases. Protein abundances are not precise estimates but indicate ranges. The range in the second column reflects the minimal and maximal protein amounts estimated across all light intensities studied in this work. Estimation of protein abundances is detailed in *Table 2—Source data 1*, a list of all proteins is provided in *Table 2—Source data 2*. The experimental conditions of (**Moal and Lagoutte, 2012**) are comparable to the conditions used in this study with the exception of high light used here and distinct *Synechocystis* substrains (**Figure 2—source data 2**).

| Protein complex | Molecules per cell | Method | Stoichiometry | Reference |
|---|---|---|---|---|
| Elongation factor | 179000–274000 | Proteomics | TufA | This study |
| Phosphoglycerate kinase | 45000–73000 | Proteomics | Pgk | This study |
| Ribosome small subunit | 36000–66000 | Proteomics | Rps1A,1B,B,C,D,E,F,G,H,I,J,K,L,M,N,O,P,Q,R,S,T,U | This study |
| Phycobilisome (phycocyanin) | 12000–23000 | Proteomics | $((CpcA,B)_{18},C1,C2,D,G)_6$ | This study |
| | 26000–66000 | Spectrophotometry | | This study |
| Photosystem I | 31000–63000 | Proteomics | $(PsaA,B,C,D,E,F,I,J,K,L,M,X)_3$ | This study |
| | 96000 | Spetroscopy | | *Keren et al., 2004* |
| | 540000 | Spetroscopy | | *Moal and Lagoutte, 2012* |
| Ribosome large subunit | 33000–54000 | Proteomics | RplA,B,C,D,E,F,I,J,K,L,M,N,O,P,Q,R,S,T,U,V,W,X,Y, RpmA,B,C,E,F,G,H,I,J | This study |
| Transketolase | 31000–50000 | Proteomics | $TktA_2$ | This study |
| PII signal transducing protein | 36000–46000 | Proteomics | $GlnB_3$ | This study |
| Photosystem II | 23000–46000 | Proteomics | $(PsbA1,A2,B,C,D,E,F,H,I,J,K,L,M,N,O,T,U,V,X,Y,Z, Ycf12)_2$ | This study |
| | 17000–29000 | Immunoblotting | | This study |
| | 100000 | Spetroscopy | | *Moal and Lagoutte, 2012* |
| RuBisCO | 26000–43000 | Proteomics | $(RbcL, RbcS)_8$ | This study |
| | 39000–63000 | Immunoblotting | | This study |
| Ferredoxin-NADP reductase (FNR) | 33000–42000 | Proteomics | PetH | This study |
| | 140000 | Immunoblotting | | *Moal and Lagoutte, 2012* |
| D-fructose 1,6-bisphosphatase class 2 | 29000–36000 | Proteomics | Slr20944 | This study |
| Phycobilisome (allophycocyanin) | 19000–38000 | Proteomics | $(ApcA,B)_{34},C_6,D_2,E_6,F_2$ | This study |
| | 9000–19000 | Spectrophotometry | | This study |
| G3P dehydrogenase | 21000–32000 | Proteomics | $Gap2_4$ | This study |
| Plastocyanin | 15000–29000 | Proteomics | PetE | This study |
| Superoxide dismutase [Fe] | 14000–25000 | Proteomics | $SodB_2$ | This study |
| Orange carotenoid protein | 15000–24000 | Proteomics | $Slr1963_2$ | This study |
| RNA polymerase | 8000–15000 | Proteomics | $RpoA_2,B,C1,C2,D,E,F$ | This study |
| Cytochrome b6/f | 8000–15000 | Proteomics | $(PetA,B,C2,D,G,L,M,N)_2$ | This study |
| Chaperonine GroEL | 7000–13000 | Proteomics | $GroL1_{14}$ | This study |
| Ribosome recycling factor | 6000–7000 | Proteomics | Frr | This study |
| Phosphoglycerate dehydrogenase | 3000–5000 | Proteomics | $SerA_4$ | This study |
| Pyruvate dehydrogenase | 3000–4000 | Proteomics | $(PdhA, PdhB)_2$ | This study |
| Glutamine synthetase | 2000–4000 | Proteomics | $GlnA_{12}$ | This study |
| Isocitrate dehydrogenase | 2000–3000 | Proteomics | $Icd_2$ | This study |

*Table 2 continued on next page*

*Table 2 continued*

| Protein complex | Molecules per cell | Method | Stoichiometry | Reference |
|---|---|---|---|---|
| Glycogen synthase | 2000–3000 | Proteomics | GlgA1 | This study |
| DNA polymerase III | 1000–2000 | Proteomics | DnaN$_2$ | This study |
| Pyruvate kinase | 1000–2000 | Proteomics | Pyk2$_4$ | This study |
| Acetyl-coenzyme A carboxylase | 1000 | Proteomics | AccB, AccC, AccA$_2$,ACCD$_2$ | This study |
| Carbonic anhydrase | 400–700 | Proteomics | IcfA$_6$ | This study |
| Acetyl-coenzyme A reductase | 300–600 | Proteomics | PhaB$_4$ | This study |
| Circadian clock proteins KaiA/ KaiB/KaiC | 200–500 | Proteomics | KaiA$_2$/KaiB$_4$/KaiC$_6$ | This study |

DOI: https://doi.org/10.7554/eLife.42508.026

The following source data is available for Table 2:

**Source data 1.** Calculations of selected protein complex copies in *Synechocystis* cells.
DOI: https://doi.org/10.7554/eLife.42508.027

**Source data 2.** List of all proteins quantified by proteomics measurements in *Synechocystis* cells.
DOI: https://doi.org/10.7554/eLife.42508.028

A sub-maximal specific growth rate in buffered BG-11 medium might also relate to the increase in glycogen content with increasing light intensity and growth rate (*Figure 2G,H*). The relative amounts of glycogen in *Synechocystis* observed in this study were well within values reported in the literature (*Figure 2—source data 2*).

However, from the perspective of optimal resource allocation, glycogen accumulation is seemingly suboptimal, since the required energy and carbon is stored and not utilized to enhance growth. Various growth limitations are known to induce accumulation of storage products, including glycogen (*Monshupanee and Incharoensakdi, 2014*), and a recent study showed that glycogen plays an important role in energy balancing and energy homeostasis in *Synechocystis* (*Cano et al., 2018*). We therefore hypothesize that the observed increase in glycogen content, in the absence of other stress factors, is consistent with a limitation in buffered BG-11 medium. This hypothesis is also supported by varying amounts of glycogen reported for the fast-growing strain *Synechococcus elongatus* UTEX 2973: while *Abernathy et al. (2017)* only report $1.5 \pm 0.5\%$ glycogen of dry weight under fastest growth conditions, *Ungerer et al. (2018)* report a drastic increase in glycogen content when entering linear growth phase, and *Tan et al. (2018)* report up to 54.9% glycogen of dry weight under high light conditions (but unknown growth rate) — suggesting that glycogen accumulation is indicative of growth limitation by other factors than light and carbon availability.

The true growth limit of *Synechocystis* (and other cyanobacteria) remains an open question. Compared to the fast growing strain *Synechococcus elongatus* UTEX 2973, the strain used in this study showed substantially lower carbon partitioning into protein content (23–40% of dry weight, compared to 50% in *Synechococcus* 2973), and increased carbon partitioning into glycogen (8.4–22.9% of dry weight, compared to 1.5% in *Synechococcus* 2973 during the fastest growth (*Abernathy et al., 2017*)). The *Synechocystis* substrain GT-L used here also maintained a lower PSI/ PSII ratio (1.35 compared to 2–3.5 in *Synechococcus* 7942 and even higher in *Synechococcus* 2973 (*Ungerer et al., 2018*)) and did not increase the amount of electron transport carriers such as plastocyanin (Kruskal-Wallis test: $p-value = 0.731$) or cytochrome $b_6f$ (Kruskal-Wallis test: $p-value = 0.493$) with increasing light intensity and growth rate. All these factors may contribute to relatively slower growth compared to the fastest growing cyanobacteria. The recent studies of *Ungerer et al. (2018)* and *Abernathy et al. (2017)* demonstrate how just a few mutations in genetically very similar strains can lead to dramatic growth differences — differences that are likely due to different, but as yet not fully understood, cellular strategies in resource allocation. We note, however, that the main goal of our study was not to maximize cyanobacterial growth per se, but to understand resource allocation in a widely used model strain.

We also note that many of the commonly used strains, including substrains of *Synechocystis* sp. PCC 6803, have been maintained in laboratories and in culture collections for extended periods of

time, and may have therefore acquired mutations that enhance viability in the lab, but concomitantly reduce maximal growth rates. Indeed, an instance where a cyanobacterial model strain appears to have lost, through laboratory domestication, behaviors that are important in a natural environment was recently reported (*Yang et al., 2018*).

## Cell morphology and variability of physiological parameters

Overall, the morphology and range of physiological data obtained in this study were in good agreement with previously published values for *Synechocystis* (see *Figure 2—figure supplement 1* and *Table 2* for detailed comparison). The cell diameter and volume (*Figure 2A,B*) were well within the range of values reported in the literature (*Lea-Smith et al., 2016*; *Zavřel et al., 2017*; *Rosana et al., 2012*). Likewise, the photosynthetic quotient PQ was well within values reported in the literature (*Zavřel et al., 2017*; *Shastri and Morgan, 2005*) and did not vary significantly with growth rate. The total protein content reported here (23 − 40% of gDW, *Figure 2I*) was lower than in several previous studies (*Touloupakis et al., 2015*; *Shastri and Morgan, 2005*).

As noted above, variability in physiological parameters observed in the literature (*Figure 2—figure supplement 1*, *Table 2*) can often be attributed to differences in cultivation setup, including selection of particular *Synechocystis* substrain (*Morris et al., 2017*; *Zavřel et al., 2017*). Additionally, the choice of analytical technique can affect the results, especially with respect to absolute quantification. We are aware of limitations of some techniques used in this work, including glycogen estimation (where the extracellular polymeric substances can potentially lead to overestimation of glycogen content), proteins extraction (where some proteins, especially those with transmembrane domains, could be potentially extracted with reduced efficiency), total protein quantification (where bovine serum albumin, used as a protein standard, does not have to represent cyanobacterial proteins properly), quantification of individual proteins (where the mass spectrometer ionization efficiency could potentially be affected for proteins with lower amount of charged amino acid), relative DNA estimation by flow cytometry (where penetration of SYBR Green I solution to the cells as well as SYBR Green I binding to both DNA and RNA could potentially differ under increasing light intensity), or phycobiliproteins determination (where the values from proteomics analysis and spectrophotometric analysis differed, *Table 2*). Nevertheless, even taken these technical limitations into account, the quantities reported here fit well into the previously reported ranges of *Synechocystis* physiology (*Figure 2—figure supplement 1*, *Table 2*), as well as to the predictions of the proteome allocation model (*Figure 6*).

## Trends in physiological parameters

Of particular interest were the trends of physiological parameters with respect to increasing light intensity and growth rate. Almost all identified parameters showed significant changes in dependence of light intensity and growth rate, including cell size (diameter and volume , *Figure 2A*), gas exchange rates (*Figure 2C–F*), as well as glycogen (*Figure 2G–H*), DNA and pigment content (*Figure 2K–N*). Trends in physiological parameters were consistent with previous studies. The increase in gas exchange ($O_2$ release and basal respiration) has been observed previously (*Zavřel et al., 2015b*; *Zavřel et al., 2017*). Likewise, the increase in cellular size with growth rate (*Figure 2A*) has been reported in *Synechocystis* (*Zavřel et al., 2017*; *Cordara et al., 2018*) as well as in heterotrophic bacteria, yeast or mammalian cells (*Aldea et al., 2017*). Light was also shown to affect DNA content (ploidy level) in *Synechocystis* (*Zerulla et al., 2016*), however, no study of DNA content change with growth rate is available to date.

Reduction of light harvesting pigments under high light is well documented in the literature (e.g. (*Zavřel et al., 2018a*; *Jahn et al., 2018*)). Interestingly, we found upregulation of chlorophyll *a*, phycobilins and both PSII and PSI proteins synthesis in *Synechocystis* cells in the initial part of the growth curve (i.e. between light intensities of 27.5 − 220 μmol(photons) $m^{-2}s^{-1}$, *Figure 2L,N*, *Figure 6C*). Similar trends have been described in *Synechocystis* (*Zavřel et al., 2017*) as well as in other cyanobacteria and algae (*Kumar et al., 2011*; *Wu et al., 2015*). Different from most previous studies, the range of light intensities tested here also included conditions of photoinhibition. In several parameters, in particular glycogen content (*Figure 2G–H*) or pigment content (*Figure 2K–N*), we observed a characteristic 'kink', that is, a sharp in- or decrease of the respective abundances.

This finding emphasizes photoinhibition as a distinct growth regime and distinguishes phototrophic growth laws from their heterotrophic counterparts.

Our findings also emphasize the need to specify to which reference value the particular changes are reported. Typically, values in the literature are reported relative to optical density as a proxy for cellular dry weight—making a direct comparison between experimental conditions difficult. Furthermore, if cellular composition or cell size changes, these changes do not necessarily translate into corresponding changes per cell or per protein content.

## Proteome allocation with growth rate

Beyond physiological parameters, we followed the global proteome allocation as a function of growth rate. The most pronounced changes in proteome with increasing light intensity and growth rate were related to upregulation of translational proteins and downregulation of photosynthetic proteins (*Table 1*, *Figure 6*). The upregulation of proteins related to translation (*Figure 6B–C*) is consistent with well-established growth laws for heterotrophic growth. In particular, *E. coli* shows consistently increased proteome investment into translation-related proteins with increasing growth rate (*Peebo et al., 2015*). Unique for photosynthetic organisms, we observed a decrease of (relative) allocation to proteins annotated with photosynthesis (*Figure 6B–C*). These results are also consistent with a recent study from *Jahn et al. (2018)*. Likewise, the observed decrease is also in agreement with predictions from resource allocation models (*Burnap, 2015*; *Faizi et al., 2018*), even for simple models that do not consider photoinhibition (*Burnap, 2015*). While the RbcL subunit of RuBisCo showed a slight increase with increasing growth rate (*Figure 6C*), we observed no general upregulation of metabolic proteins in the proteomics data with increasing growth rate (*Figure 6B*)—a deviation from the growth laws predicted by the model considered here, as well as from the growth laws predicted by other (heterotrophic) models (*Molenaar et al., 2009*). This finding indicates that the metabolic capacity itself is sufficient for high growth rates, even under conditions where lack of light input limits faster growth. We hypothesize that the most pronounced changes with changing light intensity are observed for proteins related to translation and photosynthesis due to two facts: Firstly, translation is typically limited by ribosomal capacity, requiring an upregulation of translational capacity with faster growth rates. In addition, the short half-life of the D1 protein requires the cell to adjust the translational capacity at high light intensities. Secondly, overcapacity of light harvesting may give rise to detrimental effects, such as increased cellular (photo-)damage. In comparison, overcapacity in the metabolic dark reaction does not entail obvious detrimental consequences (other than the loss of the invested resources) and therefore might be under less evolutionary pressure to change with changing light intensity. We could further corroborate this hypothesis in silico using the proteome allocation model: by artificially forcing a constant mass fraction of a proteome class, we were able to evaluate the impact of such sub-optimal acclimationon the specific growth rate as a function of light intensities. While constant mass fractions of ribosomal and photosynthetic proteins resulted in a marked deviation in the specific growth rate, a constant metabolic fraction only resulted in a minor deviation (*Figure 6—figure supplement 2*).

## Interpretation of the results in the context of a coarse-grained model

The coarse-grained model of phototrophic growth allows us to interpret the physiological and proteomic changes in the context of (optimal) protein allocation. We emphasize that the model was not constructed or parametrized to reproduce certain observed behavior – rather it represents an independent null-hypothesis that provides information about the expected changes in proteome fractions with increasing growth rate under the assumption of (evolutionary) optimality. In line with models of heterotrophic growth (*Molenaar et al., 2009*; *Weiße et al., 2015*), the model predicts an increase in allocation of ribosomal proteins as a function of growth rate (*Figure 6B–C*). Different to heterotrophic models, however, the model also predicts a characteristic upward 'kink' under conditions of photoinhibition. The relative proteomics data confirms this behavior, including the 'kink' at high light intensities (*Figure 6B–C*). The sharp upregulation of ribosomes in the model is due to the increased turnover of proteins subject to photodamage. As previously noted in *Faizi et al. (2018)*, the model is likely to overestimate this effect, due to the fact that within the model, photodamage is exclusively related to an increase in protein turnover. We expect that in *Synechocystis* also other repair mechanisms are active, resulting in a less pronounced upregulation of ribosomes and energy

usage elsewhere. Indeed, the observed upregulation in the data is less pronounced than in the model simulations (*Figure 6B–C*). Furthermore, the model predicts a downregulation of the light harvesting machinery with increasing light intensity (*Figure 5B*) and growth rates (*Figure 6B–C*). The relative proteome allocation confirms this trend, including again the predicted 'kink' when entering photoinhibition (*Figure 6B–C*). Interestingly, the characteristic 'kinks' were not observed in the recent study of *Jahn et al. (2018)* — possibly because the experimental condition used therein only considered a single light condition in the photo-inhibited growth regime.

Finally, as for models of heterotrophic growth, the model predicts an increase in the proteome fraction related to metabolic processes with increasing growth rate (*Figure 6B–C*). The metabolic proteome fraction, in particular enzymes related to a genome-scale metabolic reconstruction (*Knoop et al., 2013*), did not exhibit such a clear upregulation with the exception of the RbcL protein (a subunit of RuBisCo) that increased in relative abundance with increasing growth rate (*Figure 6C*). We note that, different from our results, the recent study of *Jahn et al. (2018)* reported an increase in the metabolic proteome fraction with increasing light intensity, albeit also less than expected compared to their computational growth model. However, a direct comparison of our results with the results of *Jahn et al. (2018)* is challenging, due to differences in definition of the respective enzyme classes: the proteome fraction corresponding to the metabolic enzyme ($M$) class within our analysis was assigned using the metabolic reconstruction of *Knoop et al., 2013*, and consists only of the respective metabolic enzymes (excluding transporters and the electron transport chain). The metabolic enzyme class of *Jahn et al. (2018)* follows a manually curated definition based on annotation from Cyanobase and is considerably broader (the Ci uptake, fixation, and metabolism class (CBM) also includes proteins annotated with translation and other processes). Furthermore, we note that the fold changes of proteins within the $M$-fraction are inhomogenous: as can be observed in *Figure 3B*, proteins associated with biosynthesis and small molecule metabolic processes exhibit up- as well as down-regulation as a function of growth rate—a fact that is also reflected in the proteins of the $M$-fraction.

## Conclusions

Despite the importance of cyanobacteria as photosynthetic model organisms and as host organisms for green biotechnology, as yet only few studies have addressed quantitative growth properties and resource allocation even for well characterized model strains. The goal of this study was therefore to close this gap with respect to knowledge and interpretation of key physiological parameters of the cyanobacterial model strain *Synechocystis* sp. PCC 6803 in dependence of light intensity and growth rate. We focused on light as the only variable environmental parameter – and identified trends in key physiological parameters and proteome allocation as a function of growth rate. The interpretation of data was facilitated by a coarse-grained computational model of cyanobacterial resource allocation and the data was put into the context of data available in the literature, obtained by a comprehensive literature research. Overall, the resulting growth laws (decrease of proteome fraction associated with light harvesting and increase of proteome fraction associated with translation with increasing light intensity and growth rate) are in good agreement with previous theoretical (*Burnap, 2015*; *Faizi et al., 2018*) and experimental studies (*Jahn et al., 2018*), whereas the observed invariance of the proteome fraction associated with metabolic processes differed from model predictions.

Light, however, is not the only factor that affects photoautotrophic growth. Further studies are required to identify growth limitation under different environmental conditions, in particular limitations induced by other biotechnologically or environmentally relevant macro- or micronutrients. Ultimately, such studies will also have to take into account the diversity of cyanobacterial metabolism (*Beck et al., 2018*). As indicated by rather minor genetic differences between strains with vastly different growth rates, we expect that differences in many biotechnologically relevant parameters between strains are indeed a consequence of different strategies in resource allocation — making further studies of cellular accounting a key prerequisite for successful green biotechnology. The proposed reproducible cultivation setup and the coarse-grained computational model used in this study provide a suitable framework and reference to facilitate and to contribute to such studies.

## Materials and methods

**Key resources table**

| Reagent type (species) or resource | Designation | Source or reference | Identifiers | Additional information |
|---|---|---|---|---|
| Antibody | Rabbit Anti-PsbA | Agrisera | Cat. #: AS05 084 RRID:AB_2172617 | WB (1:10000) |
| Antibody | Rabbit Anti-PsaC | Agrisera | Cat. #: AS10 939 RRID:AB_10752085 | WB (1:1000) |
| Antibody | Rabbit Anti-RbcL | Agrisera | Cat. #: AS03 037 RRID:AB_2175406 | WB (1:5000) |
| Antibody | Rabbit Anti-S1 | Agrisera | Cat. #: AS08 309 RRID:AB_1271140 | WB (1:2000) |
| Antibody | Rabbit Anti-L1 | Agrisera | Cat. #: AS11 1738 RRID:AB_10754471 | WB (1:1000) |
| Peptide, recombinant protein | Recombinant PsbA from *Synechocystis* sp. PCC 6803 | Agrisera | Cat. #: AS01 016S | 41.5 kDa |
| Peptide, recombinant protein | Recombinant PsaC from *Synechocystis* sp. PCC 6803 | Agrisera | Cat. #: AS04 042S | 11.5 kDa |
| Peptide, recombinant protein | Puri1ed spinach RbcL | Agrisera | Cat. #: AS01 017S | 52.7 kDa |

### Inoculum cultures

*Synechocystis* sp. PCC 6803 GT-L was obtained from Prof. D. A. Los (Timiryazev Institute of Plant Physiology, Moscow, RU). The strain was cultivated in BG-11 medium (*Stanier et al., 1971*) supplemented with 17 mM HEPES (Carl Roth, Karlsruhe, Germany, $pK_a$ = 7.5). pH of the buffered BG-11 was adjusted to 8.2. The inoculum cultures were precultivated in 250 mL Erlenmeyer flasks on a standard orbital shaker (120 rpm) in a cultivation chamber tempered at 25°C under an average illumination of 110 μmol(photons) $m^{-2}s^{-1}$ (provided by cool white light LEDs) and under 1% $CO_2$ in the atmosphere.

### Photobioreactor

Growth experiments were performed in flat panel photobioreactors, described in detail previously (*Nedbal et al., 2008*). The illumination in the photobioreactors was designed as a chessboard configuration of red and blue LEDs (red: $\lambda_{max} \approx 633$ nm, $\lambda_{1/2} \approx 20$ nm, Luxeon LXHLPD09; blue: $\lambda_{max} \approx 445$ nm, $\lambda_{1/2} \approx 20$ nm, Luxeon LXHL-PR09; all manufactured by Future Lighting Solutions, Montreal, QC, Canada). Spectral characteristics of the LEDs are shown in *Zavřel et al., 2015b*. The photobioreactor continuously measured optical density (OD) by an inbuilt densitometer and steady-state pigment fluorescence emission yield by an inbuilt fluorometer (both described in *Nedbal et al., 2008*). Dissolved $O_2$ was monitored by the InPro6800 electrode, culture temperature and pH were monitored by the InPro3253 electrode (all manufactured by Mettler-Toledo Inc, Columbus, OH, USA). Culture homogenization was secured by the inflow gas bubbling with a rate of 200 mL $min^{-1}$, complemented by rotations of a magnetic stirrer bar ($\phi 5 \times 35$ mm, 210 rpm) in a vertical plane. All other photobioreactor accessories were the same as described in (*Zavřel et al., 2015b*). The photobioreactor setup is visualized in *Figure 1A*.

### Experimental setup

Growth characterization was performed in a quasi-continuous regime as described previously (*Zavřel et al., 2015b*). Briefly, the exponentially growing *Synechocystis* cells were maintained in a defined range of optical density (measured at 680 nm, $OD_{680}$) by controlled dilution of the culture suspension with fresh buffered BG-11 medium (turbidostat). The optical density was measured by the photobioreactor instrument base, and the $OD_{680}$ range was set to 0.60–0.66, which corresponded to approximately $2 - 4 \times 10^7$ cells $mL^{-1}$. Starting $OD_{680}$ of all cultures was 0.1 - 0.2, which corresponded to approximately $2 - 4 \times 10^6$ cells $mL^{-1}$. Once the culture density reached $OD_{680}$ 0.66, the quasi-continuous cultivation setup was initiated by starting automated cultures dilution within the selected $OD_{680}$ range. Under each light condition, the cultures were growing for at least 24 hr.

This period was long enough to reach growth stability, that is to acclimate the cells to the specific condition. The principal of quasi-continuous cultivation is represented in *Figure 1B*.

During the quasi-continuous experiments, *Synechocystis* was cultivated under red light intensities of 27.5 – 1000 µmol(photons) m$^{-2}$s$^{-1}$. The cultures were always supplemented with low intensity of blue light (27.5 µmol(photons) m$^{-2}$s$^{-1}$) in order to avoid growth limitation by complete absence of short wavelength photons (*Golden, 1995*). Cultivation temperature was set to 32°C, and the experiments were performed under a $CO_2$ concentration of 5000 ppm in the sparging gas (secured by the Gas Mixing System GMS 150, Photon System Instruments Ltd., Brno, CZ).

## Analytical methods

### Growth rates determination

Specific growth rates µ were evaluated from an increase of $OD_{680}$ signal as recorded by the photobioreactor during the quasi-continuous cultivation (after the growth stabilized under each particular light intensity), according to *Zavřel et al. (2015b)*:

$$\mu = \frac{ln\frac{OD_{680\ t_2}}{OD_{680\ t_1}}}{t_2 - t_1}, \tag{1}$$

where $OD_{680\ t_1}$ and $OD_{680\ t_2}$ represent optical densities measured at 680 nm in times $t_1$ and $t_2$, respectively.

As an alternative method, specific growth rates were determined from depletion of spare cultivation medium, as measured by top loading balances (Ind231, Mettler-Toledo Inc, Columbus, OH, USA, *Figure 1C*), according to the following equation:

$$\mu = \frac{f}{V}, \tag{2}$$

where $f$ represents average flow rate of spare cultivation medium and $V$ represents volume of the culture suspension in the photobioreactor.

### Determination of photosynthesis and respiration rates

The oxygen evolution rates as a sum of all oxygen fluxes between *Synechocystis* cells and cultivation media (net photosynthesis, NP) and dark respiration rates (R) were determined from the signal of $dO_2$ electrode in the photobioreactor vessel by turning off aeration for 10 min, through 5 min light and 5 min dark periods, according to *Červený et al., 2009*. Gross photosynthesis rates (rates of oxygen production by water splitting, GP) were calculated as: GP = NP + R (photorespiration and other processes were neglected for the GP calculations).

Carbon uptake (net $CO_2$ uptake rate as a sum of all $CO_2$ fluxes between *Synechocystis* cells and cultivation media) was determined from the steady-state values of $CO_2$ concentration in the photobioreactor output gas, as measured by the Gas Analyzing System (Photon System Instruments Ltd., Brno, CZ, described in detail in *Červený et al., 2009*).

### Pigment content measurements

Content of chlorophyll $a$, carotenoids and phycobilisomes was measured spectrophotometrically following the protocols of *Zavřel et al. (2015a)* and *Zavřel et al. (2018a)*.

### Measurements of glycogen, cell size and DNA content

Content of glycogen was measured spectrophotometrically, following the protocol of *Zavřel et al. (2018b)*. Cellular dry weight was measured using XA105DR analytical balances (Mettler Tolledo, Greifensee, CH). Cell count was measured with the Cellometer Auto M10 (Nexcelom Bioscience, Lawrence, MA, USA).

Cell size was determined using the ImageStream MkII imaging flow cytometer (Amnis Corp., Seattle, WA, USA). Right after harvesting from the photobioreactor, 500 µL of the culture suspension was centrifuged (4 000 g, 4 min, 25C), supernatant was discarded, pellet was resuspended in 0.25% glutaraldehyde solution and the samples were incubated for 10 min at laboratory temperature. The fixed cells were stored in -80°C until further processing (up to 2 months in total). For further analysis, the samples were thawed on ice for 2 hr, and they were kept at laboratory temperature in dark for

additional 30 min after thawing (after 20 min, 5 µL of SYBR Green I solution was added to each sample for DNA content estimation; for details see the next paragraph). During the cytometric analysis, only bright field images were collected by the imaging flow cytometer. Gating of the measured populations was applied to discriminate: a) focused objects (using combination of both RMS gradient and Treshold Mask features of IDEAS software), and b) round objects (width/length ratio between 0.9 – 1.0). The imaging flow cytometer was calibrated with non-fluorescent microspheres (1 – 15 µm, Thermo Fisher Scientific, Waltham MA, USA) and the results were validated with the light microscope Axio Imager 2 (Carl Zeiss, Oberkochen, DE). During the cytometric analysis, also chlorophyll fluorescence (excitation: 488 nm, detection: 480 - 560 nm) and phycobilisomes fluorescence (excitation: 642 nm, detection: 642 nm - 745 nm) were measured to validate selection of the cells within all measured objects.

DNA content was measured in the same samples as the cell size. After the samples thawing on ice for 2 hr and at laboratory temperature for 20 min (see the previous paragraph for details), 5 µL of SYBR Green I solution (Thermo Fisher Scientific, Waltham, MA USA, diluted 1:100 in DMSO) was added to 500 µL of the culture suspension to mark cellular DNA, and the samples were further incubated for 10 min in dark at laboratory temperature. During the cytometric analysis, a 488 nm argon laser was used to excite both SYBR Green I and chlorophyll $a$, and another 642 nm laser was used to excite phycobilisomes. To identify *Synechocystis* cells within all measured objects, the same gating as described in the previous paragraph was used.

## Protein extraction

Protein extraction was performed according to *Brown et al. (2008)* with modifications. For each sample, 90 mL of the culture suspension was withdrawn from the photobioreactor, centrifuged (4 000 xg, 5 min, 32°C), supernatant was partially discarded (leaving 0.5–1 ml of liquid in the original 50 mL conical tube) and the pellet was resuspended and transferred to 1.5 mL Eppendorf tube. The tubes were centrifuged (20 000 x g, 4 min, 32°C), supernatants were completely discarded and the tubes were stored at −80°C until further processing (up to 4 months). All following steps of protein isolation were performed at 4°C. The frozen pellets were resuspended in 0.8 mL of a protein extraction buffer (50 mM Tris-HCl (pH 7.6); 2 mM EDTA; 10 mM $MgCl_2$; 250 mM sucrose, 1% of protease inhibitor cocktail P9599, Sigma-Aldrich, St. Louis, MO, USA). The mixture was transferred to 2 mL tubes with a rubber o-ring (containing 0.5 mL of sand and glass beads) and the cells were disrupted by 6 × 30 s homogenization pulses on the laboratory mixer (BeadBug Microtube Homogenizer, Benchmark Scientific, Sayreville, NJ, USA). Between each pulse, the samples were kept on ice. After the first step of homogenization, the samples were shortly centrifuged, 200 µL of 10% SDS was added to each tube (to reach the final concentration of 2%), and the samples were mixed and frozen in liquid nitrogen. Right after freezing, the cells were additionally sonicated in an ultrasound bath with ice until thawing (six cycles, between each cycle the samples were frozen in liquid nitrogen). After ultrasound homogenization, the samples were centrifuged (10 000 x g, 3 min, 4°C) to remove unbroken cells and cell debris, and 500 µL of the supernatant protein fraction was transferred to a new 1.5 mL Eppendorf tube. The total protein concentration was measured in triplicates with a bicinchoninic acid assay kit (BCA1-1KT, Sigma-Aldrich, USA) by the method of *Smith et al. (1985)* using bovine serum albumin (A7906, Sigma-Aldrich, USA) as a standard. The samples were used for both immunoblotting and proteomics measurements.

## Immunoblotting protein analysis

Immunoblotting and protein quantification was done according to *Brown et al. (2008)* with modifications. 100 µl of each sample was diluted with equal volume of 2x loading buffer (100 mM Tris-HCl (pH 7.6); 20 mM DTT, 4% SDS 0.02% bromphenol blue, 20% glycerol), denatured for 20 min at 37°C and centrifuged (10 000 x g, 20 min, laboratory temperature) before loading. Samples containing 4 µg of total protein were separated in 12.5% (for detection of RbcL, S1, L1) or 15% (for detection of D1, PsaC) 0.75 mm thick polyacrylamide mini gels by SDS-PAGE at 200 V for 40–50 min in a MiniProtean Tetra Cell (Bio-Rad, Hercules, CA, USA). Separated proteins were transferred to 45 µm nitrocellulose membranes (Hybond-C Extra, GE Healthcare Life Sciences, Chicago, Il, USA) using the Trans-Blot Turbo Transfer system (BioRad, Hercules, CA, USA) at 25 V, 1.0 A, laboratory temperature, and cycle duration of 30 min. The nitrocellulose membranes were blocked immediately after transfer in

TBST-G buffer (10 mM Tris-HCl (pH7.6); 150 mM NaCl; 0.05% (v/v) Tween-20; 1% cold-water fish gelatin) for 2 hr at laboratory temperature. Primary antibodies diluted in TBST-G buffer were used according to recommendations of the manufacturer. The list of primary antibodies is provided in *Figure 6—figure supplement 1*. After incubation of the membranes in the primary antibody solutions for 1 hr at laboratory temperature, the solutions were poured off and the membranes were briefly rinsed and washed 3 times for 15 min in TBST buffer at laboratory temperature. For signal detection, the membranes were incubated with goat anti-rabbit immunoglobulin G horseradish peroxidase conjugated antibodies diluted 1:75000 in TBST buffer for 1 hr at laboratory temperature. Membranes were washed as described above and developed with Clarity Western ECL Substrate (Bio-Rad, Hercules, CA, USA) according to the manufacturer's instructions. Images of the blots were obtained using a Gel Doc XR + system (Bio-Rad, Hercules, CA, USA).

Intensity of protein bands on immunoblots was estimated by densitometric analysis with the Image Lab 5.1 software (Bio-Rad, Hercules, CA, USA). The protein concentrations were quantified as relative to the lowest light intensity (27.5 $\mu$mol(photons) m$^{-2}$s$^{-1}$). In addition, absolute amounts of PsbA, PsaC, and RbcL proteins were estimated from standard curves prepared by serial dilutions of corresponding standard proteins. The list of protein standards is provided in *Figure 6—figure supplement 1*.

## Quantitative proteomics

Protein lysates of 5 individually grown replicate samples per group (27.5-55-110-220-440-1100 $\mu$mol (photons) m$^{-2}$s$^{-1}$) were prepared for mass spectrometric analysis by shortly stacking 5 $\mu$g proteins per sample in a 4 - 12% Bis-Tris sodium dodecyl sulfate (SDS)-polyacrylamide gel (Thermo Scientific, Darmstadt, Germany) over a 4 mm running distance. Proteins were further processed as described previously (*Poschmann et al., 2014*). Briefly, gels were subjected to a silver staining protein containing bands cut out from the gel, destained, washed, reduced with dithiothreitol and alkylated with iodoacetamide. Subsequently, proteins were digested for 16 hr at 37°C with 0.1 $\mu$g trypsin (Serva, Heidelberg, Germany), peptides were extracted from the gel and after drying in a vacuum concentrator resuspended in 0.1% trifluoroacetic acid. 500 ng of sobulized peptides per sample were then analyzed by a liquid chromatography (Ultimate 3000 Rapid Separation Liquid Chromatography system, RSLC, Thermo Fisher Scientific, Dreieich, Germany) coupled with quantitative mass spectrometry. First, peptides were loaded for 10 min at a flow rate of 6 $\mu$l/min on a trap column (Acclaim PepMap100 trap column, 3 $\mu$m C18 particle size, 100 Å pore size, 75 $\mu$m inner diameter, 2 cm length, Thermo Fisher Scientific, Dreieich, Germany) using 0.1 % trifluoroacetic acid as mobile phase. Subsequently, peptides were separated at 60°C on an analytical column (Acclaim PepMapRSLC, 2 $\mu$m C18 particle size, 100 Å pore size, 75 $\mu$m inner diameter, 25 cm length, Thermo Scientific, Dreieich, Germany) at a flow rate of 300 nl/min using a 2 hr gradient from 4 to 40% solvent B (solvent A: 0.1% (v/v) formic acid in water, solvent B: 0.1% (v/v) formic acid, 84% (v/v) acetonitrile in water).

Separated peptides were injected via distal coated SilicaTip emitters (New Objective, Woburn, MA, USA) into a Q Exactive plus Orbitrap mass spectrometer (Thermo Fisher Scientific, Dreieich, Germany) online coupled via a nanosource electrospray interface. The mass spectrometer was operated in data dependent positive mode with a capillary temperature of 250°C and spray voltage set to 1 400 V. First, full scans were recorded in profile mode at a resolution of 70,000 over a scan range from 350 to 2 000 m/z. Ions were accumulated for a maximum of 80 ms and the target value for automatic gain control was set to 3,000,000. Second, a maximum of ten two- or threefold charged precursor ions were selected within a 2 m/z window using the build in quadrupole, fragmented via higher-energy collisional dissociation and fragments analyzed in the Orbitrap over a maximal scan range from 200 to 2 000 m/z at a resolution of 17,500. Here, the automatic gain control was set to 100,000 and the maximum ion time was 60 ms. For the next 100 s already fragmented precursors were excluded from further analysis.

## Peptide and protein identification

For peptide and protein identification and quantification the MaxQuant software suite (version 1.6.1.0, MPI for Biochemistry, Planegg, Germany) was used with standard parameters if not otherwise stated. For database searches 3507 protein entries from the UP000001425 *Synechocystis* sp.

strain PCC 6803 downloaded on the 20th of November 2017 from the UniProtKB were considered. Searches were conducted using following parameters: carbamidomethylation at cysteines as fixed and oxidation at methionine and N-terminal protein acetylation as variable modification, false discovery rate on peptide and protein level 1%, match between runs enabled as well as label-free quantification and iBAQ, tryptic cleavage specificity with a maximum of two missed cleavage sites. A first search was conducted with a precursor mass tolerance of 20 ppm and after recalibration by Max-Quant, 4.5 ppm precursor mass tolerances were applied. The mass tolerances for fragment spectra signals were set to 20 ppm.

Quantitative information for identified proteins was further processed within the Perseus framework (version 1.6.1.1, MPI for Biochemistry, Planegg, Germany). Here, only non-contaminant proteins identified with at least two different peptides were considered. Additionally, all proteins were filtered out which - in at least one group – did not show any missing values in the label-free quantification data which then was used after log2 transformation for statistical analysis and relative protein amount comparisons between the different light intensity groups. Calculations of protein stoichiometries and comparison to quantitative protein data derived from other methods was done on absolute quantitative data based on iBAQ intensities. First, iBAQ intensities were normalized on the sum iBAQ intensities of four proteins (Q55806, P72587, P73505, Q59978) showing a small standard deviation, similar intensity range and ratio close to one between the mean intensities of the 27.5 and 1100 µmol(photons) $m^{-2}s^{-1}$ group. Second, a calibration of absolute intensities was performed using the PsaC Western blot data (mean of 104 fmol/µl). The mass spectrometry proteomics data have been deposited to the ProteomeXchange Consortium via the PRIDE (*Vizcaíno et al., 2016*) partner repository with the dataset identifier PXD009626.

## Proteomaps

For generating proteomaps, the version 1.0 of the visualization tool at www.proteomaps.net (*Liebermeister et al., 2014*) was used, choosing absolute quantitative values and *Synechocystis* sp. 6803 as organism. To be compatible with the proteomaps tool, the mass spectrometric data was searched against the 3661 entries from the GCA_000009725.1 protein dataset from CyanoBase downloaded on 22th January 2018.

## Statistical analysis
### Kruskal-Wallis test

For the identification of cellular resources that significantly changed with growth rate (including each single protein out of total 1356 identified proteins), we performed a Kruskal-Wallis test (Python scipy.stats module) for each resource (null hypothesis was that the median of all compared groups is equal) and did a pair-by-pair comparison of two conditions in each case. For the test we compared only those measurements with at least three samples. Cellular components and proteins determined as significantly changing with light intensity and growth rate were those that had at least one pair that differed significantly with a $p-value < 0.05$.

### Fisher's exact test

We further performed a Fisher's exact test to investigate which of the GO categories filtered out from the proteomics dataset are significantly associated to growth related proteins. For this test we used the GO slim categories. Therefore, we classified the 1356 proteins into growth dependent (779 proteins) and independent groups (577 proteins). The second classification criterion referred to being in one specific gene ontology group or not. The test was then performed for each GO slim category. An imbalance for one GO slim category, between the amount of growth-dependent and growth-independent proteins, was determined as significant for a $p-value < 0.05$.

## A coarse-grained proteome allocation model
### Model overview

The previously published model of proteome allocation of *Faizi et al. (2018)* was extended with a growth-independent protein class Q that accounts for approximately half of the proteome. The growth-dependent proteome is comprised of transporter (T), ribosomes (R), metabolic enzymes (M) and photosynthetic units (P). Furthermore, protein degradation and an energy maintenance term

were added, resulting in a basal energy expenditure. A description of the modified model with all reaction rates and parameters is provided in *Figure 5* and *Supplementary file 1*.

The proteome allocation model gives rise to an optimization problem. We assume that the objective of a unicellular organism is to maximize its growth rate while the proteome mass remains constant. The maximization of the cellular growth rate, for a given external condition, is achieved by re-adjusting the amount of ribosomes that are delegated to translate a specific protein. The optimization problem was solved using the APMonitor Optimization Suite (*Hedengren et al., 2014*) with the steady-state optimization mode and the IPOPT (Interior Point Optimizer) solver option. The python interface was used to run the model.

## Model parametrization and fitting

The model describes growth per cellular dry weight. Cell size only affects the estimated parameter for diffusion of inorganic carbon. For simplicity, the diffusion parameter is set constant (with a cell diameter of approximately 2 μM). Parameters were as in *Faizi et al. (2018)* and sourced from the primary literature (*Mangan and Brenner, 2014*; *Marcus et al., 2005*; *Dornmair et al., 1989*; *Omata et al., 2002*; *Bremer and Dennis, 2008*; *Maier et al., 2011*; *Knoop et al., 2013*). Only three parameters $\tau$ (turnover rate of the photosynthetic unit), $k_d$ (photodamage) and $\sigma$ (effective absorption cross-section) were then fitted to the measured growth rates. No protein data were used in the fitting. Parameter estimation was done for an external inorganic carbon concentration of $c_i^x$ = 100 mM ($c_i^x$ saturated condition). To minimize the computational effort, a pre-defined set of values for these parameters was specified prior to fitting,

$$\tau = \{50, 75, 100\}, \tag{3}$$

$$k_d = \{5 \cdot 10^{-7}, 6 \cdot 10^{-7}, ..., 4 \cdot 10^{-6}, 5 \cdot 10^{-6}\}, \tag{4}$$

$$\sigma = \{0.1, 0.2, ..., 0.1\}. \tag{5}$$

To select the best fit, the negative logarithm of the likelihood was calculated for each parameter set:

$$l(\theta) = \sum_i \frac{(y_i(\theta) - x_i)^2}{e_i^2} + log(2 \cdot \pi \cdot e_i^2), \tag{6}$$

where $x_i$ represents the here measured growth rates with their uncertainties $e_i$ and $y_i(\theta)$ the simulated growth rates calculated with the model parameters $\theta$. The best fit $l(\theta)$ = -51.46, was obtained with $\tau$ = 75 s$^{-1}$, $k_d$ = 10$^{-6}$ and $\sigma$ = 0.7 nm$^2$. Compared to the original model, the addition of the growth-independent protein fraction enhances the energy demand of the cell, and increases the turnover rate and absorption cross-section of the photosystem. We emphasize that the purpose of the model was not to provide an exact fit to the data, but to guide the interpretation of the results.

## Impact of non-adaptive protein fractions on the estimated growth rate

To investigate the potential influence of a constant (non-adaptive) protein mass fraction of Ribosome, Photosynthetic unit, and Metabolic proteins classes (as shown in *Figure 6*) on the predicted growth rate, an additional constraint was added to the optimization problem, such that the concentration of the respective protein class is

$$[Z] = \frac{\varphi_Z \cdot D_c}{n_Z}, \tag{7}$$

where $D_c$ is the cell density (in units of amino acids per cell), $n_Z$ determines the length of the enzyme Z, and $\varphi_Z$ is the (constant) mass fraction of the protein class Z. In addition, to account for the fact that proteins can be de- or activated (by post-translational modifications such as phophorylation), an additional variable $\alpha_Z$ was introduced that determines the amount of active enzymes (such that the amount of catalytically active enzyme $Z_a$ is $[Z_a] = \alpha_Z \cdot [Z]$). The growth rate is then optimized using the remaining protein classes, as well as the parameter $\alpha_Z$ as variables. The value for the constant protein fraction was set such that it corresponds to the mass fraction of the respective protein class at the highest growth rate.

## Acknowledgments

TZ and JČ were supported by the Ministry of Education, Youth and Sports of the Czech Republic within the National Sustainability Program I (NPU I), grant number LO1415, under OP RDE grant number CZ.02.1.01/0.0/0.0/16–026/0008413 'Strategic Partnership for Environmental Technologies and Energy Production', and by GA CR, Grant number 18–24397S. Access to instruments and facilities was supported by the Czech research infrastructure for systems biology C4SYS (project no. LM2015055). MF was supported by the German Research Foundation (DFG), Research Training Group 1772/2 (Computational Systems Biology). AZ and MS were supported by the Russian Science Foundation, grant number 14-14-00904. GP and KS were supported by the German Research Foundation (DFG), CRC1208 'Identity and Dynamics of Membrane Systems - from Molecules to Cellular Functions. RS was funded by the grant 'CyanoGrowth' of the German Federal Ministry of Education and Research as part of the 'e:Bio -Innovationswettbewerb Systembiologie' [e:Bio - systems biology innovation competition] initiative (reference: FKZ 0316192).

## Additional information

### Funding

| Funder | Grant reference number | Author |
|---|---|---|
| Ministerstvo Školství, Mládeže a Tělovýchovy | CZ.02.1.01/0.0/0.0/16–026/0008413 | Tomáš Zavřel Jan Červený |
| Grantová Agentura České Republiky | 18–24397S | Tomáš Zavřel Jan Červený |
| Deutsche Forschungsgemeinschaft | | Marjan Faizi |
| Deutsche Forschungsgemeinschaft | CRC1208 | Gereon Poschmann Kai Stühler |
| Russian Science Foundation | 14-14-00904 | Maria Sinetova Anna Zorina |
| Bundesministerium für Bildung und Forschung | FKZ 0316192 | Ralf Steuer |

The funders had no role in study design, data collection and interpretation, or the decision to submit the work for publication.

### Author contributions

Tomáš Zavřel, Conceptualization, Data curation, Formal analysis, Investigation, Visualization, Methodology, Writing—original draft, Project administration, Writing—review and editing, Designed all experiments, Carried out all aspects of experiments and collected the data, Measured and analyzed morphology and physiology data, Wrote the paper; Marjan Faizi, Data curation, Formal analysis, Investigation, Visualization, Writing—review and editing, Created the computational model, Analyzed proteomics data, Wrote the paper; Cristina Loureiro, Formal analysis, Investigation, Carried out all aspects of experiments and collected the data; Gereon Poschmann, Data curation, Methodology, Measured and analyzed the proteomics data, Assisted with preparing the manuscript; Kai Stühler, Data curation, Methodology, Assisted with proteomics data measurement; Maria Sinetova, Anna Zorina, Data curation, Investigation, Methodology, Analyzed protein content in cells, Performed the immunoblotting analysis, Assisted with preparing the manuscript; Ralf Steuer, Conceptualization, Supervision, Investigation, Writing—original draft, Writing—review and editing, Wrote the paper, Supervised the project, Supervised the computational model preparation and analysis of proteomics data; Jan Červený, Conceptualization, Supervision, Funding acquisition, Project administration, Writing—review and editing, Designed all experiments, Supervised the project, Assisted with preparing the manuscript

Author ORCIDs

Tomáš Zavřel (ID) https://orcid.org/0000-0002-0849-3503

Marjan Faizi (ID) http://orcid.org/0000-0002-7170-7762

Cristina Loureiro (ID) http://orcid.org/0000-0002-9245-2447

Gereon Poschmann (ID) http://orcid.org/0000-0003-2448-0611

Ralf Steuer (ID) http://orcid.org/0000-0003-2217-1655

Jan Červený (ID) http://orcid.org/0000-0002-5046-3105

Decision letter and Author response
Decision letter https://doi.org/10.7554/eLife.42508.034
Author response https://doi.org/10.7554/eLife.42508.035

## Additional files

### Supplementary files

• Supplementary file 1. Summary of the proteome allocatioin model.
DOI: https://doi.org/10.7554/eLife.42508.029

• Transparent reporting form
DOI: https://doi.org/10.7554/eLife.42508.030

### Data availability

Proteomics data have been deposited to the ProteomeXchange Consortium under accession code PXD009626.

The following dataset was generated:

| Author(s) | Year | Dataset title | Dataset URL | Database and Identifier |
|---|---|---|---|---|
| Poschmann G | 2018 | Synechocystis sp. proteome on different light conditions | https://www.ebi.ac.uk/pride/archive/projects/PXD009626 | ProteomeXchange, PXD009626 |

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
