## [Decision Letter]

Thank you for submitting your article "Quantitative insights into the cyanobacterial cell economy" for consideration by *eLife*. Your article has been reviewed by three peer reviewers, including Severin Sasso as the Reviewing Editor and Reviewer #1, and the evaluation has been overseen by Ian Baldwin as the Senior Editor. The following individual involved in review of your submission has agreed to reveal his identity: Jörg Toepel (Reviewer #3).

The reviewers have discussed the reviews with one another and the Reviewing Editor has drafted this decision to help you prepare a revised submission.

Summary:

Zavrel et al. present a comprehensive analysis of the growth of the model cyanobacterium, *Synechocystis* sp. PCC 6803. It is one of the first extensive studies of cyanobacterial growth that inventories nearly all of the key material components, including the proteome, glycogen, chlorophyll *a* and carotenoids. This manuscript also evaluates the most critical physiological parameters of growth under varying conditions, such as growth rate, cell volume, and photosynthesis and respiration rates. This is being done under different conditions of illumination ranging from light-limiting to photoinhibitory light levels. Importantly, the results are interpreted in the context of a model of phototrophic growth that has intrinsic predictive power independent of parameters obtained from the modeled experiments. While specific aspects of this work were previously published by other authors (e.g. Jahn et al., 2018; Cordara et al., 2018), the manuscript by Zavrel is unique bringing all of the analyses together in a single cultivation setup (along with a sophisticated model). In addition, large amounts of literature values are comprehensively summarized so that as a whole, this manuscript is suitable for *eLife*'s Tools and Resources section.

Essential revisions:

1) While carotenoids were measured (Figure 2—source data 1), other lipids such as glycerolipids were not determined. If frozen cell pellets from the same experiment are still available, the total lipid content should be quantified.

2) There is a recent article also studying quantitative growth under turbidostat conditions (Jahn et al., 2018). This study reports similar growth rates, but appears not to be as comprehensive in terms of physiological data collected and uses a more simplified model. The authors should comment on this article in terms of the results and modeling and compare their proteomics data to those published by Jahn et al.

3) The relatively slow growth of *Synechocystis* in comparison to other model cyanobacteria is interesting, and although this does not diminish the value of the outstanding current findings, it will be important to broaden the discussion. Specifically, the observation of the accumulation of glycogen to higher levels than might be expected, has a good treatment at the beginning of the Discussion section, but could be usefully extended. For example, the hypothesis that the growth media may still not be optimal is reasonable. However, recent findings with closely related strains of *Synechococcus elongatus* indicate that growth differences and specifically, a slowdown associated with additional glycogen accumulation, can be due to intrinsic genetic differences (Ungerer et al., 2018, studying these *Synechococcus* strains). Those studies showed that just a handful of mutations produced the dramatic growth differences that appear to be due to changes in the expression of key proteins of the light and dark reactions of photosynthesis among other changes in protein expression and enhance anabolic capacity based upon metabolomic analysis (Abernathy et al., 2017). This observation of very different growth rates between genetic very similar strains of *Synechococcus* comes from the realization that many of the strains that have been maintained in labs and culture collections may acquire mutations that enhance viability in the lab, but may also reduce maximal growth rates.

4) In some of the figures, it is hard to assign individual data points to one of the three stages (photolimitation, photosaturation or photoinhibition). The authors should consider using three different colors for visualization, one color for each stage, particularly in Figures 2 and 6.

5) Large parts of the model of phototrophic growth (Figure 5) were previously published by Faizi et al., 2018. This should be noted in the main text or Figure 5, too, together with a short explanation of how the model was extended. The authors should also explain how the model provides new information in this study compared to the previous publication.

6) The authors can state more clearly what the substantial gain of knowledge of their work is (cf. Summary above). For example, a few sentences could be added to the manuscript's Conclusions.

---

## [Author Response]

Essential revisions:1) While carotenoids were measured (Figure 2—source data 1), other lipids such as glycerolipids were not determined. If frozen cell pellets from the same experiment are still available, the total lipid content should be quantified.

Unfortunately, no frozen cell pellets from the same experiments are still available. Therefore the analysis as suggested by the reviewer is not possible.

2) There is a recent article also studying quantitative growth under turbidostat conditions (Jahn et al., 2018). This study reports similar growth rates, but appears not to be as comprehensive in terms of physiological data collected and uses a more simplified model. The authors should comment on this article in terms of the results and modeling and compare their proteomics data to those published by Jahn et al.

Indeed, the results reported in the recent article of Jahn et al., 2018 (which was not yet available during preparation of our manuscript) are closely related to our study – albeit not as comprehensive in terms of physiological data.

Most of the overall results with respect to proteome allocation are mutually supportive. A main difference is that Jahn et al. do not seem to observe the characteristic ‘kink’ in several physiological parameters and proteome fractions under conditions of photoinhibition – possibly due to the fact that in their work only a single datapoint under conditions of photoinhibition was considered.

With respect to the computational model, a major difference is that the model of Jahn et al. predominantly relies on fitting (in particular proteome allocation under changing light intensities. Thus, strictly speaking, the model-derived growth rate in their Figure 3B should not be called a ‘model prediction’). In contrast, our model has only 3 free parameters which were adjusted to match the experimental growth curve: within our model, these three parameters correspond to (i) setting the maximal growth rate, (ii) setting the strength of photoinhibition (the slope of the decrease in growth rate as a function of light intensity), and (iii) setting the units of the light intensity (via the effective cross-section, which provides a proportionality factor between the incoming and the absorbed light). No proteome data was used during parametrization. We therefore consider our model to be a true prediction in the sense that it serves as an independent hypothesis about optimal proteome allocation.

We modified our manuscript in several places to take into account the new reference and to comment on the differences.

Introduction:

– We now cite in the Introduction: “In contrast, only few studies so far have addressed the limits of cyanobacterial growth from an experimental perspective (Bernstein et al., 2016; Yu et al., 2015; Abernathy et al., 2017; Ungerer et al., 2018; Jahn et al., 2018)”.

Results:

– The results of Jahn et al., 2018 are now included in the comparison of quantitative data reported in the literature (Figure 2—figure supplement 2), in section “Maximum specific growth rate” (comparison).

Discussion:

Quantitative resource allocation in cyanobacteria

– Citation added: “Cyanobacteria are increasingly important host organisms for green biotechnology, but as yet insight into resource allocation of these organisms is restricted to few studies (Abernathy et al., 2017; Burnap, 2015; Faizi et al., 2018; Jahn et al., 2018).”

Maximal growth rates and glycogen accumulation

– Citation added: “The maximal specific growth rates of *Synechocystis* GT-L obtained in this study (Figure 1C, D) were similar to the maximal growth rates of other *Synechocystis* substrains reported in previous studies (Touloupakis et al., 2015; Nguyen and Rittmann, 2016; Du et al., 2016; Jahn et al., 2018).”

Proteome allocation with growth rate:

– “The most pronounced changes in proteome with increasing light intensity and growth rate were related to upregulation of translational proteins and downregulation of photosynthetic proteins (Table 1, Figure 6). […]These results are also consistent with a recent study from Jahn et al., 2018.”

Interpretation of results in the context of a coarse-grained computational model:

– “Interestingly, the characteristic 'kinks' were not observed in the recent study of Jahn et al. (2018) – possibly because the experimental condition used therein only considered a single light condition in the photo-inhibited growth regime.”

– “We note that, different from our results, the recent study of Jahn et al. (2018)reported an increase in the metabolic proteome fraction with increasing light intensity, albeit also less than expected compared to the computational growth model.”

Conclusions:

– “Overall, the resulting growth laws (decrease of proteome fraction associated with light harvesting and increase of proteome fraction associated with translation and biosynthesis with increasing light intensity and growth rate) is in good agreement with previous theoretical (Burnap 2015; Faizi et al., 2018) and experimental studies (Jahn et al., 2018).”

3) The relatively slow growth of Synechocystis in comparison to other model cyanobacteria is interesting, and although this does not diminish the value of the outstanding current findings, it will be important to broaden the discussion. Specifically, the observation of the accumulation of glycogen to higher levels than might be expected, has a good treatment at the beginning of the Discussion section, but could be usefully extended. For example, the hypothesis that the growth media may still not be optimal is reasonable. However, recent findings with closely related strains of Synechococcus elongatus indicate that growth differences and specifically, a slowdown associated with additional glycogen accumulation, can be due to intrinsic genetic differences (Ungerer et al., 2018, studying these Synechococcus strains). Those studies showed that just a handful of mutations produced the dramatic growth differences that appear to be due to changes in the expression of key proteins of the light and dark reactions of photosynthesis among other changes in protein expression and enhance anabolic capacity based upon metabolomic analysis (Abernathy et al., 2017). This observation of very different growth rates between genetic very similar strains of Synechococcus comes from the realization that many of the strains that have been maintained in labs and culture collections may acquire mutations that enhance viability in the lab, but may also reduce maximal growth rates.

We fully agree. While our primary aim was to provide an in-depth analysis of physiology and proteome allocation in a widely-used model strain, we now broadened the discussion about maximal growth rates of cyanobacteria. In particular, we added a new paragraph to the section “Maximal growth rates and glycogen accumulation” taking the comments of the reviewer into account:

“The true growth limit of Synechocystis (and other cyanobacteria) remains an open question. […] Indeed, an instance where a cyanobacterial model strain appears to have lost, through laboratory domestication, behaviors that are important in a natural environment was recently reported (Yang et al., 2018).”

4) In some of the figures, it is hard to assign individual data points to one of the three stages (photolimitation, photosaturation or photoinhibition). The authors should consider using three different colors for visualization, one color for each stage, particularly in Figures 2 and 6.

We fully agree. Figure 2 and Figure 6B-C now contain three different levels of shading to distinguish between photolimitation, photosaturation and photoinhibition stages, as suggested by the reviewer. The captions of Figure 2 and Figure 6 were modified accordingly:

“Within each figure, data points are displayed in three different color shades to reflect (from bright to dark) light-limited, light-saturated and light-inhibited growth.”

In line with this, we also modified Figure 5B by adding the corresponding proteome allocation as a function of light intensity, where the distinction between the three light regimes is easier to observe. We also modified the last sentence of the caption of Figure 5 as follows “The model reproduces the measured growth curve […] Shown are the specific growth rate µ, as well as the main proteome fractions, ribosome R, photosynthetic electron transport P, and metabolism M, as a function of light intensity.

Furthermore, we refer to Figure 5B in the second paragraph of the section “A coarse-grained model provides insight into proteome allocation”.

5) Large parts of the model of phototrophic growth (Figure 5) were previously published by Faizi et al., 2018. This should be noted in the main text or Figure 5, too, together with a short explanation of how the model was extended. The authors should also explain how the model provides new information in this study compared to the previous publication.

We reformulated the first sentence of the legend in Figure 5 to ‘A coarse-grained model of phototrophic growth, adopted from Faizi et al., 2018.’

To better detail the modifications of the model, we added and extended the following paragraph in the section “A coarse-grained model provides insight into proteome allocation”:

“Compared to the original model from Faizi et al., 2018, we now include a growth-independent protein fraction *Q* that accounts for half of the proteome mass. […] No protein data were used during model parametrization and fitting.”

Corresponding descriptions were also added to the Materials and methods section:

Model overview:

– “Furthermore, protein degradation and an energy maintenance term were added, resulting in a basal energy expenditure.”

Model parametrization and fitting:

– “Compared to the original model, the addition of the growth-independent protein fraction enhances the energy demand of the cell, and increases the turnover rate and absorption cross-section of the photosystem. We emphasize that the purpose of the model was not to provide an exact fit to the data, but to guide the interpretation of the results.”

We note that, overall, the model as such was not intended to provide new information on its own (as compared to the original model). Rather its strength is that it serves as an independent guide (a hypothesis) for the interpretation of our results. We deliberately chose not to extensively modify the model compared to the previous publication, and, in particular, not to use the newly acquired data obtained in our study to constrain or otherwise “train” (using the terminology of Jahn et al.) the model and its results.

To further demonstrate the utility of the model, however, we have now also extended the computational analysis. In particular, the model allowed us to test the consequences of non-adaptive protein allocation (i.e. a scenario where a certain protein fraction is artificially held constant across many light intensities, and only the remaining protein fractions are allowed to adapt to maximize the growth rate). We tested the impact of a “clamped” protein mass fraction for the enzyme classes R (ribosomes), M (metabolism), and P (photosynthesis). The results show that “clamping” the metabolic enzyme fraction has the least overall impact on growth rate (in accordance with the fact that the metabolic fraction does only exhibit slight variations as a function of light-limited growth-rate).

These results are now shown in Figure 6—figure supplement 2, a description has been added to Materials and methods subsection ‘Impact of non-adaptive protein fractions on the estimated growth rate’, and the results are discussed in the context of the less pronounced changes in protein allocation for metabolic enzymes on subsection ‘Proteome allocation with growth rate’ in the Discussion, where we now write:

“We can further corroborate this hypothesis in silico using the proteome allocation model: by artificially forcing a constant mass fraction of a proteome class, we are able to evaluate the impact of such sub-optimal adaptation on the specific growth rate as a function of light intensities. While constant mass fractions of ribosomal and photosynthetic proteins results in a marked deviation in the specific growth rate, a constant metabolic fraction only results in a minor deviation (Figure 6—figure supplement 2).”

6) The authors can state more clearly what the substantial gain of knowledge of their work is (cf. Summary above). For example, a few sentences could be added to the manuscript's Conclusions.

As stated by the reviewers, we consider the main advance of our work to bring together a comprehensive physiological and proteomic analysis of a widely used cyanobacterial model strain using a single and reproducible cultivation setup, together with a predictive model as a guide for the analysis and a comprehensive review of literature values. In particular, we also expect that differences in resource allocation underlies many of the observed differences in biotechnologically relevant parameters – making ‘cellular accounting’ and studies such as ours paramount for a successful green biotechnology.

To state these results and hypotheses more clearly, we have modified the section “Conclusions” accordingly:

“Despite the importance of cyanobacteria as photosynthetic model organisms and as host organisms for green biotechnology, as yet only few studies have addressed quantitative growth properties and resource allocation even for well characterized model strains. […] The proposed reproducible cultivation setup and the coarse-grained computational model used in this study provide a suitable framework and reference to facilitate and to contribute to such studies.”

In addition, the section “Quantitative resource allocation in cyanobacteria” (in Discussion) has been modified to better highlight some of the current challenges in understanding cyanobacterial growth:

“Cyanobacteria are becoming increasingly important as host organisms for synthesis of renewable bioproducts from atmospheric CO_2_, but as yet insight into resource allocation of these organisms has been scarce (Jahn et al., 2018; Abernathy et al., 2017; Burnap, 2015; Faizi et al., 2018). […] The results, interpreted in the context of a coarse-grained computational model of cyanobacterial resource allocation, provide further understanding of cellular economics during light-limited, light-saturated and light-inhibited cyanobacterial growth.”